# Dual targeted extracellular vesicles regulate oncogenic genes in advanced pancreatic cancer

Chi-Ling Chiang[1,2,17], Yifan Ma[1,3,17], Ya-Chin Hou[4,5,17], Junjie Pan[1], Sin-Yu Chen[6], Ming-Hsien Chien[7], Zhi-Xuan Zhang[6], Wei-Hsiang Hsu[6], Xinyu Wang[1], Jingjing Zhang[1], Hong Li[1], Lili Sun[8], Shannon Fallen[9], Inyoul Lee[9], Xing-Yu Chen[10], Yeh-Shiu Chu[10], Chi Zhang[11], Tai-Shan Cheng[6], Wen Jiang [12], Betty Y. S. Kim [13], Eduardo Reategui[1], Robert Lee[11], Yuan Yuan[1,8], Hsiao-Chun Liu[4,5], Kai Wang[9], Michael Hsiao [7], Chi-Ying F. Huang [6] ✉, Yan-Shen Shan [4,5] ✉, Andrew S. Lee [14,15] ✉ & L. James Lee [1,6,16] ✉

Pancreatic ductal adenocarcinoma (PDAC) tumours carry multiple gene mutations and respond poorly to treatments. There is currently an unmet need for drug carriers that can deliver multiple gene cargoes to target high solid tumour burden like PDAC. Here, we report a dual targeted extracellular vesicle (dtEV) carrying high loads of therapeutic RNA that effectively suppresses large PDAC tumours in mice. The EV surface contains a CD64 protein that has a tissue targeting peptide and a humanized monoclonal antibody. Cells sequentially transfected with plasmid DNAs encoding for the RNA and protein of interest by Transwell®-based asymmetric cell electroporation release abundant targeted EVs with high RNA loading. Together with a low dose chemotherapy drug, Gemcitabine, dtEVs suppress large orthotopic PANC-1 and patient derived xenograft tumours and metastasis in mice and extended animal survival. Our work presents a clinically accessible and scalable way to produce abundant EVs for delivering multiple gene cargoes to large solid tumours.

Pancreatic ductal adenocarcinoma (PDAC) is difficult to treat and has a low 5-year survival rate of <10%[1]. GTPase KRAS mutations occur in >90% of PDAC[2] and the only clinically available drugs targeting the RAS family are inhibitors for KRAS$^{G12C}$ mutation in non-small cell lung cancer (LUMAKRAS™, sotorasib)[3]. Most patients at the time of diagnosis have advanced unresectable tumors that carry multiple gene mutations[4]. In 70% of these cases, KRAS$^{G12D}$ mutations co-occur with mutated TP53 tumor suppressor[5]. Therefore, there is a need for new treatments that target multiple mutations in PDAC. However, delivering multiple genetic agents precisely to such advanced tumors is a significant challenge. Liposomal nanoparticles (LNPs), while popular, are unsuitable for solid tumor treatments because they do not

penetrate tissues or traverse physiological barriers well[6], while viral deliver efficiently but are highly immunogenic[7].

Extracellular vehicles (EVs) such as exosomes and microvesicles are attractive alternatives for gene delivery because they have low immunogenicity and cytotoxicity and can cross physiological barriers[8–10]. Exosomes containing exogenous silencing RNA for KRAS$^{G12D}$ (siKRAS$^{G12D}$) have been shown to suppress small, non-advanced, orthotopic PANC-1 PDAC tumors in mouse models[11]. We have also previously used EVs carrying endogenous phosphatase and tensin homolog (PTEN) mRNA to slow down the growth of small tumors in glioblastoma mice[12]. While promising, for these EVs-based therapies to be clinically relevant, they must be able to carry multiple

gene cargoes, precisely target, penetrate, and treat large solid PDAC tumors and metastases. Furthermore, EV production must be scalable and readily available. Current methods involve either inserting synthetic RNAs into pre-isolated EVs[11,13], which is difficult to load large mRNA cargoes, or genetically modifying donor cells[14] which are complicated and face the risk of off-target genomic integration.

In this work, we report easily scalable, dual-targeted therapeutic EVs (dtEVs) containing high copy numbers of TP53 mRNA or siKRAS[G12D] that can suppress large solid PDAC tumors (Fig. 1a). Our dtEV surface contains a CD64 (Fc-gamma receptor 1) protein engineered at the N-terminus with a CKAAKNK (CK) tissue homing peptide that targets pancreatic tumor tissue[15]. This engineered protein (CD64[ck]), which acts as a universal anchor, binds with high affinity to any clinically available therapeutic humanized monoclonal antibodies (hmAb)[16] to form a second targeting ligand on the EV surface. We use humanized anti-receptor tyrosine kinase-like orphan receptor 1 (αROR1, clone: 2A2) antibodies as the second ligand to target ROR1 receptors commonly found on tumors but not normal tissues[17]. Plasmid DNAs to produce CD64[ck] protein and either TP53 mRNA or siKRAS[G12D] are sequentially delivered into mouse embryonic fibroblast (MEF) cells or human bone marrow stem cells (hBMSCs) by Transwell®-based asymmetric cell electroporation (TACE)—a clinically accessible and scalable method using affordable Transwell® inserts. Sequentially delivering the CD64[ck] plasmid DNA and the TP53 or siKRAS[G12D] plasmid DNA by TACE promote the release of abundant EVs with high TP53 mRNA inside the EVs and CD64[ck] protein on the EV surface. After affinity binding with humanized αROR1 antibodies, our dtEVs together with a first-line chemotherapy drug, Gemcitabine (GEM), suppress large orthotopic PANC-1 and patient-derived xenograft (PDX) tumors and metastasis in mice and extended animal survival through strong gene regulation, cancer cell cycle arrest, and reduced chemodrug resistance. Our cellular trafficking results suggest that dual targeting promote receptor-mediated tissue penetration, tumor cell uptake, and cytosol RNA release of EVs. Our work demonstrates a simple, low-cost way to produce abundant targeted EVs carrying high loads of genetic cargoes that can effectively treat advanced cancer in animal models.

## Results

### Design of dual-targeted extracellular vesicles

To achieve dtEVs carrying TP53 mRNA and siKRAS[G12D], we transfected MEF or hBMSC donor cells with wild-type CD64 (CD64[wt]) or CD64[ck] (with CKAAKNK peptide), TP53, and small hairpin KRAS[G12D] (shKRAS[G12D]) containing plasmids by TACE[11] (Fig. 1a). A monolayer of cells was grown on the membrane surface of the Transwell® insert and DNA plasmids were loaded below the membrane. This setup topologically divides the cell surface into a small transfection site equivalent to the 1 μm Transwell® membrane pore beneath the cell and a large EV secretion area covering the rest of the cell surface. Electric pulses delivered to the Transwell® insert drive the negatively charged plasmids into the cells electrophoretically and non-endocytically via the Transwell® pores. Non-endocytic delivery of plasmids by TACE is more efficient than conventional electroporation (bulk electroporation, BEP) because BEP mainly creates pores on cell membrane to facilitate plasmid-membrane binding, but plasmids still need to be up-taken via endocytosis[18]. We measured human CD64 protein expression in MEF cells at different time points (6, 12, 18, and 24 h) following either BEP or TACE transfection. The electroporated cells exhibited CD64 expression as early as 6 h after TACE electroporation, while it took over 18 h for stable CD64 expression by BEP (Supplementary Fig. 1).

Because electric field strength is much lower outside the pores than inside (Supplementary Note 1), cell damage at the secretion site is mild, allowing abundant EVs to be secreted. After TACE, we measured the release of EVs and vesicular RNA or protein of interest every 4 h for 24 h. The EVs collected over 24 h were purified by size exclusion chromatography (qEV columns) and characterized. Compared to

untreated MEFs, TACE produced >30-fold EVs, with a maximum of ~$1 \times 10^6$ EVs/cell around 16 h (Fig. 1b). CD64[ck] proteins on the EV surface was ~28 molecules/EV on average and >40 molecules/EV at the peak between 12 and 16 h. Meanwhile, there were ~4 copies of TP53 mRNA per EV on average and >8 copies/EV at the peak of around 4 h (Fig. 1c). Antisense siKRAS[G12D] level was ~1000 copies/EV on average and reached a maximum of ~1900 copies/EV at ~16 h. No hairpin-type preliminary shKRAS[G12D] was detected in the EVs. Quantification of each cargo in dtEVs was performed by using calibration curves with recombinant protein of CD64, synthetic TP53 mRNA and siRNA of KRAS[G12D] as standards. The cargoes per EV was estimated by the total amount of each molecule divided by the dtEV number (Supplementary Fig. 2). The anchor protein of CD64[wt] or CD64[ck] were observed in both exosome and MV fractions which allows the preloaded human IgG (hIgG) for further targeting purposes (Fig. 1d).

We next confirmed that there were ~11% full-length TP53 mRNA in the total EV RNA obtained from TACE-stimulated MEFs by Bioanalyzer electrophoresis with synthetic TP53 mRNA as standard (Fig. 1e). Since the endogenous mRNAs have longer poly-A tails with various lengths, there is a shift in size between synthetic- and EV-mRNA. Moreover, the integrity of vesicular TP53 mRNA was also confirmed by oligo-dT primed quantitative polymerase chain reaction (qPCR) from both the 5′ end (exons 2 and 3) and 3′ end (exons 10 and 11) of the transcript from two edges of TP53 mRNA (Fig. 1f). By using two independent methods, we confirmed that the integrity of encapsulated mRNA of interest. Current methods of cellular transfection[12] or RNA insertion to EVs[13] by conventional electroporation and cellular modification[14] also increase the EV secretion and RNA loading, but the efficacy is poor with low integrity of RNA cargos[12]. They are not suitable for clinical uses.

### Sequential TACE alters donor cells for dtEV production and RNA loading

Even though CD64[ck] protein and RNA cargos were both detected in TACE-produced EVs, we observed their expressional peaks were quite different (Fig. 1b, c). The exosomal TP53 mRNA reached peak around 4 h, while the expression of CD64[ck] protein and siKRAS[G12D] required longer time for translation and maturation after 12 h of TACE. Here, we introduced the sequential TACE (sTACE) by delivering CD64[ck] plasmid followed by TP53 plasmid with an 8, 16, or 24 h gap between the two plasmids in MEF (Fig. 2a) and BMSC (Supplementary Fig. 3). The optimized sTACE with 16 h gap gave the best coordination of surface CD64[ck] protein, TP53 mRNA loading, and EV number for our the dtEV formulation. CD64[ck] and shKRAS[G12D] plasmids could be delivered together since their production profiles were similar. Since dtEVs are comprised of surface CD64[ck] and RNA cargos, we further used immunofluorescence (anti-CD64) and RNA in-situ hybridization (RNA-ISH) to measure each composition within the precursor of dtEVs, late-endosomal MVBs, in donor cells. Through the proportion of colocalized versus total fluorescent signal, we estimated that ~45% MVBs contained CD64[ck] protein and ~93% MVBs contained TP53 mRNA at 4 h, while ~55% MVBs contained siKRAS[G12D] at 12 h after TACE (Fig. 2b). The sequence and specificity of the probes and antibodies used for TP53 mRNA, siKRAS[G12D], CD64 protein, and Rab7 is shown in Supplementary Fig 4.

We further characterized single dtEVs by using an immune lipoplex nanoparticle (ILN)-total internal reflection fluorescence microscopy (TIRFM) biochip assay[19]. Individual dtEV was captured by antibodies of exosomal markers (CD9 and CD63) tethered on the chip surface, and the TP53 mRNA or siKRAS[G12D] in these captured exosomes are detected by florescent probes encapsulated in immune lipoplex nanoparticles (ILNs)[20] (Supplementary Fig. 5). Fluorescent signals from TIRFM were used to quantify the RNA content and its colocalization with anti-CD64 signal. The colocalization ratios of encapsulated TP53 mRNA or siKRAS[G12D] and CD64[ck] surface protein within dtEV were ~60% (Fig. 2c). The specificity of probes used in ILN for TP53 mRNA, and

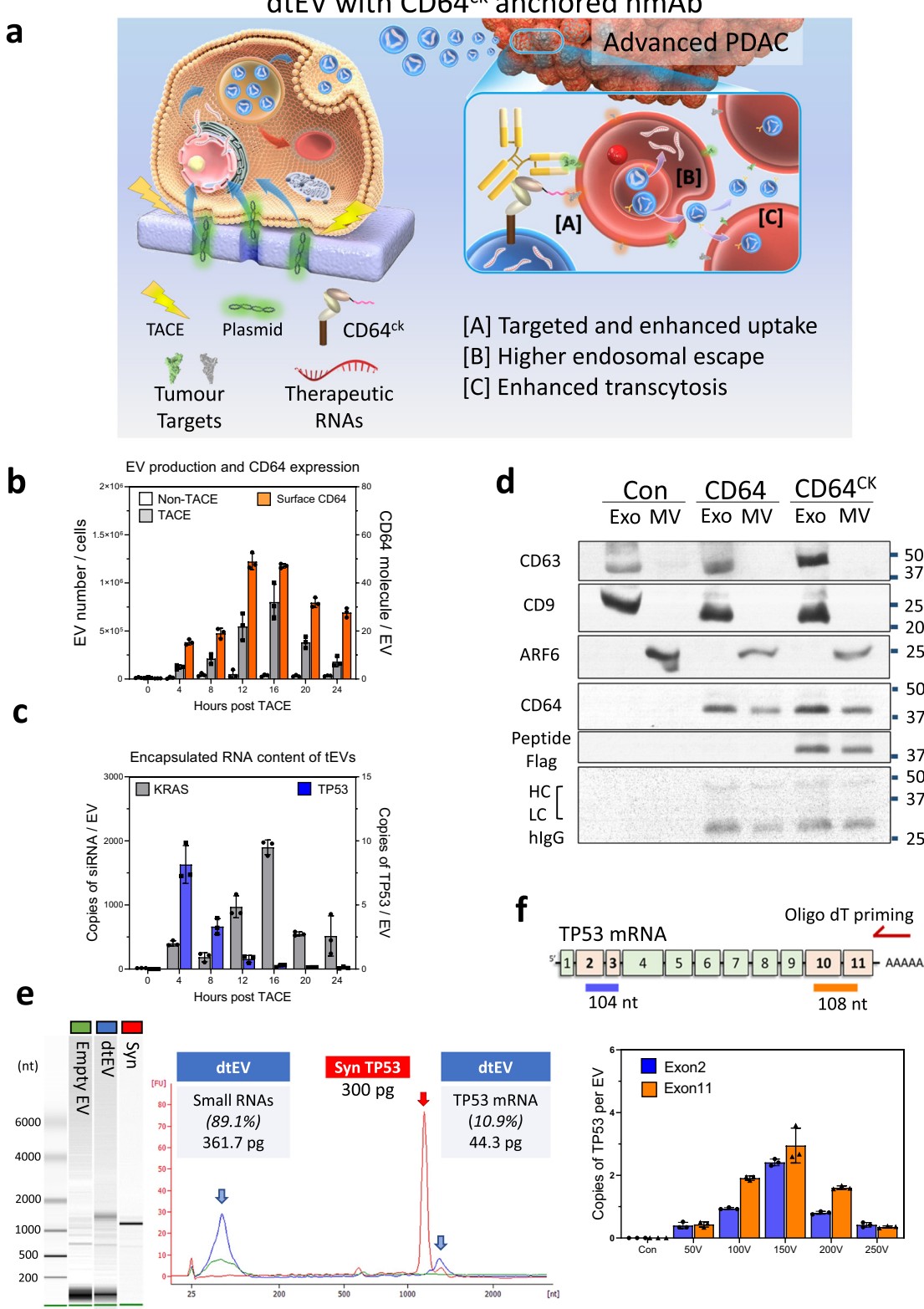

**a** dtEV with CD64^ck anchored hmAb

Advanced PDAC

[A] Targeted and enhanced uptake
[B] Higher endosomal escape
[C] Enhanced transcytosis

TACE   Plasmid   CD64^ck

Tumour Targets   Therapeutic RNAs

**b** EV production and CD64 expression

**c** Encapsulated RNA content of tEVs

**d**

**e**

**f** TP53 mRNA    Oligo dT priming

siKRAS^G12D is demonstrated in Supplementary Fig. 6. To further realize the distribution of RNA cargos (i.e. TP53 mRNA or siKRAS^G12D) in entire vesicular fraction, harvested dtEVs were sorted by size extrusion chromatography (qEV columns, IZON) into 20 fractions in which the EV number, averaged size, surface protein markers, and loaded RNA/protein cargo in each fraction were characterized. By real-time PCR analysis for RNA cargos from each fraction, TP53 mRNA was located mainly in exosomal fractions (11–14; high CD63/CD9 expression; Fig. 2d). Interestingly, siKRAS^G12D was detected in both exosomes and microvesicles, with more copies in MVs (5000–6000 copies/MV, size: 275.2–461.3 nm) that have a larger averaged volume than exosomes (150-250 copies/exosome, size: 61.8–118.0 nm).

Global RNA analysis revealed ~2700 genes and >130 microRNAs with altered concentrations in EVs from hBMSCs and MEFs electroporated with TP53 plasmids by TACE (Supplementary Table 1–2, Supplementary Data 1–2, and Supplementary Figs. 7–9). The trend in both

**Fig. 1 | Production and characterization of dtEVs. a** Donor cells on the Transwell® insert are transfected with plasmid DNA from below the insert through high-electric field strength in Transwell® pores (1 μm). Cargo RNAs transcribed from the plasmids reside inside intraluminal vesicles (ILVs) within multivesicular bodies (MVBs). TACE promotes the secretion of dtEVs containing therapeutic RNAs and CD64[ck] protein. Clinically available humanized monoclonal antibody (hmAb) anchors on CD64[ck] as the second targeting ligand. The hmAb and CK peptide dual targeted EVs (dtEVs) provide [A] specific tumor targeting and affinity-enhanced cancer cell uptake (better endocytosis), [B] improved cytosol release of therapeutic RNAs (higher endosome escape), and [C] superior tissue penetration (stronger transcytosis) by CD64[ck]/hmAb interactions with receptors on the cancer cell membrane. **b** MEFs electroporated at 150 V by TACE secreted >30-fold EVs than untreated native MEFs. EV secretion peaked at ~16 h was synchronized with CD64[ck] expression on dtEVs surface (27.6 ± 1.9 molecules/EV on average for >24 h, $n = 3$ biological independent experiments). **c**, TP53 mRNA expression in EVs peaked within 4 h (8.2 ± 1.5 copies/EV, $n = 3$ biological independent experiments) while siKRAS[G12D] expression peaked much later at 16 h (1,899.9 ± 114.5 copies/EV, $n = 3$ biological independent experiments). **d** EVs with CD64[wt] or CD64[ck] harvested from donor cells were preloaded with human IgGs and pulled down for Western blotting. Captured human IgG was on exosomes (Exo) and microvesicles (MV). Con: untreated MEFs. **e** Bioanalyzer electrophoresis with a synthetic TP53 mRNA reference confirmed the integrity and quantity of TP53 mRNA in dtEVs. The mass ratio of small RNAs vs. mRNA in dtEVs was ~9:1. **f** The integrity of vesicular TP53 mRNA was also confirmed by oligo-dT primed qPCR from the two edges of transcript as the 5' end (exons 2 and 3, blue) and 3' end (exons 10 and 11, orange) ($n = 3$ biological independent experiments). Data were presented as mean ± SD. Source data are provided as a Source data file. **d**, **e** Electrophoresis and western blot images are representative of three independent experiments.

cell types is similar. Most of the EVs containing transcripts were probably fragments (Fig. 1e), and their levels were much lower than the TP53 mRNA level encoded by the introduced plasmid. Further, because concentrations of the affected microRNAs were also much lower than the siKRAS[G12D] level, it is unlikely any individual mRNA or microRNA in the EVs will perturb the functions of the recipient cells. However, it is worth examining the collective impact of all native microRNAs in the EVs in the future because their overall concentration is higher than the non-native siKRAS[G12D].

### dtEVs provide strong cancer cell uptake and penetration ability

The dual targeting ability of dtEVs consists of preloaded humanized antibody and additional tissue homing peptides (CK) cloned into the N-terminus of CD64. Using enzyme-linked immunoassay (ELISA) with immobilized human IgG (hIgG) to pull down CD64, we first demonstrated that engineered CD64[ck] bound to hIgG1 with the same high affinity ($K_d$) as CD64[wt] (Fig. 3a). Also, flow cytometry showed that our engineered CD64[ck] is expressed on the surface of CD63-expressed dtEVs and bound well to hIgG (Fig. 3b). In PANC-1 cells expressing high levels of epidermal growth factor receptor (EGFR) and ROR1 that can be targeted with humanized anti-EGFR (αEGFR, Cetuximab, clone: C225) or anti-ROR1 (αROR1, clone: 2A2)[21]. Internalization assay using fluorescently labeled EVs revealed better cell uptake with αROR1-targeted EVs than αEGFR-targeted EVs or normal IgG control (Fig. 3c). Adding CK peptides onto the EV surface nearly doubled the uptake of αROR1-targeted EVs in PANC-1 cells. Since IgG exists in healthy human serum at 6–16 g/L[22], we evaluated the stability of preloaded humanized antibodies on EVs in 50% human serum for 6 h at 37 °C. No significant loss in targeting ability was seen with EVs carrying either αROR1 or αEGFR (Supplementary Fig. 10a).

Transcytosis of extracellular vesicles[23] or antibodies[24,25] was previously reported, but whether they could pass the cellular barrier as an entire complex remains unclear. The putative transcytosis of dtEVs was determined by two (top and bottom) monolayers of PANC-1 cells separated by a Transwell® membrane with 5 μm pores (Fig. 3d). Fluorescent EVs and LNPs were applied to the top monolayer (1st recipient, >90% confluence) and their presence in the bottom monolayer (2nd recipient) was measured after 24 h. Adding hmAb on the EV surface enhanced transcytosis by 1–4-fold, with dtEV bearing CD64[ck] and αROR1 (CD64[ck]_αROR1) performing the best (Fig. 3e). Compared to LNPs, 2–3-fold more dtEVs were found in the bottom cell monolayer (Supplementary Fig. 10b).

Enhanced endocytosis of dtEVs was observed in single-cell level as well. The faster binding and uptake occurred with CD64[ck]_ αROR1 dtEVs (mean uptake time: 600.2 ± 31.4 s) than with single-targeted EVs (stEVs) bearing CD64[ck] alone (805.6 ± 47.5 s) or non-targeted LNPs (956.2 ± 51.4 s) (Supplementary Fig. 11). More counts of dtEVs (18.4 ± 9.5 vesicles/cell) than stEVs (13.0 ± 8.7 vesicles/cell) and LNPs (15.1 ± 12.2 particles/cell) were taken by PANC-1 cells after 45 min

(Supplementary Fig. 12). The enhanced uptake of dtEVs was mainly mediated by clathrin- and caveolae-relative endocytosis. Inhibiting clathrin- (Pitstop 10 mM)[26] and caveolae- (Methyl-β-cyclodextrin, 10 mM)[27] mediated endocytosis, exosomal secretion (neticonazole, 10 mM)[28], and intracellular trafficking (Brefeldin A 5 μM, and Exo1 20 μM) in the first recipient cells prevented putative transcytosis of dtEVs (Fig. 3f).

The strong penetration of dtEVs was noted in PANC-1 spheroids (~300 μm) as well. dtEVs penetrated deeper into the core of tumor spheroids than single targeting bearing either αROR1 or CK alone after 24 h of incubation, while LNPs did not penetrate to the core of spheroids (Fig. 3g). A similar trend was seen at 6 h where dtEVs bearing CD64[ck] and αEGFR could target the spheroids well, but the dtEVs bearing CD64[ck] and αROR1 showed better internalization in individual cells (Supplementary Fig. 13).

### Trafficking of dtEVs in recipient cells

The process of drug delivery within tissues and cells is highly complex. In this study, we present experimental results aimed at investigating the distinctions in cellular trafficking between dtEVs and other nanocarriers. Typically, recipient cells take up nanocarriers through endocytosis, leading to intracellular trafficking from the endosome to the lysosome (Fig. 4a). We treated PANC-1 cells with fluorescent dtEVs, non-targeted IgG-EVs, or LNPs (indicated by red) for 2 h. Subsequently, the cells were fixed and co-stained with anti-Rab5 to visualize the extent of colocalization with early endosomes (indicated by green). We observed similar levels of encapsulation in early endosomes of PANC-1 cells for all three nanocarriers (Fig. 4b, co-localization % (Col%): LNP: 89.0 ± 3.5; non-targeted EV: 88.17 ± 5.2; dtEV: 81.50 ± 3.7, $n = 3$). In contrast, the levels of colocalization (indicated by yellow) among the three nanocarriers in lysosomes exhibited significant variations, with dtEVs showing the lowest level in lysosomes (green) (Fig. 4c, Col%: LNP: 26.17 ± 4.0; IgG-EV: 13.53 ± 4.87; dtEV: 5.73 ± 1.45, $n = 3$). These findings suggest a potentially superior ability of dtEVs to escape the endosomal compartment. However, further rigorous experimental analyses are required to confirm this observation. While the exact mechanism of endosomal escape of EVs remains unclear, there is growing evidence suggesting that the release of EV lumens differs from that of conventional liposomal and polymeric nanoparticles[29,30]. To delve deeper into this phenomenon, we utilized a cytosol fluobody recognition system[29] to monitor the interaction of dtEV lumens throughout vesicular trafficking (Fig. 4d). Prior to the experiment, recipient cells were transfected with an anti-GFP fluobody tagged with mCherry (indicated by red), which exhibits widespread expression within the cytosol (see plasmid construct in Supplementary Fig. 14). Following the release of the dtEV lumen, the cytosolic anti-GFP fluobody exhibited recognition of CD63[GFP] releasing from the dtEV lumens, leading to a transient local accumulation. PANC-1 cells expressing anti-GFP mCherry fluobody were incubated with dtEVs containing CD63[GFP] for 2, 4, and 8 h. We observed gradual increase in the accumulation of

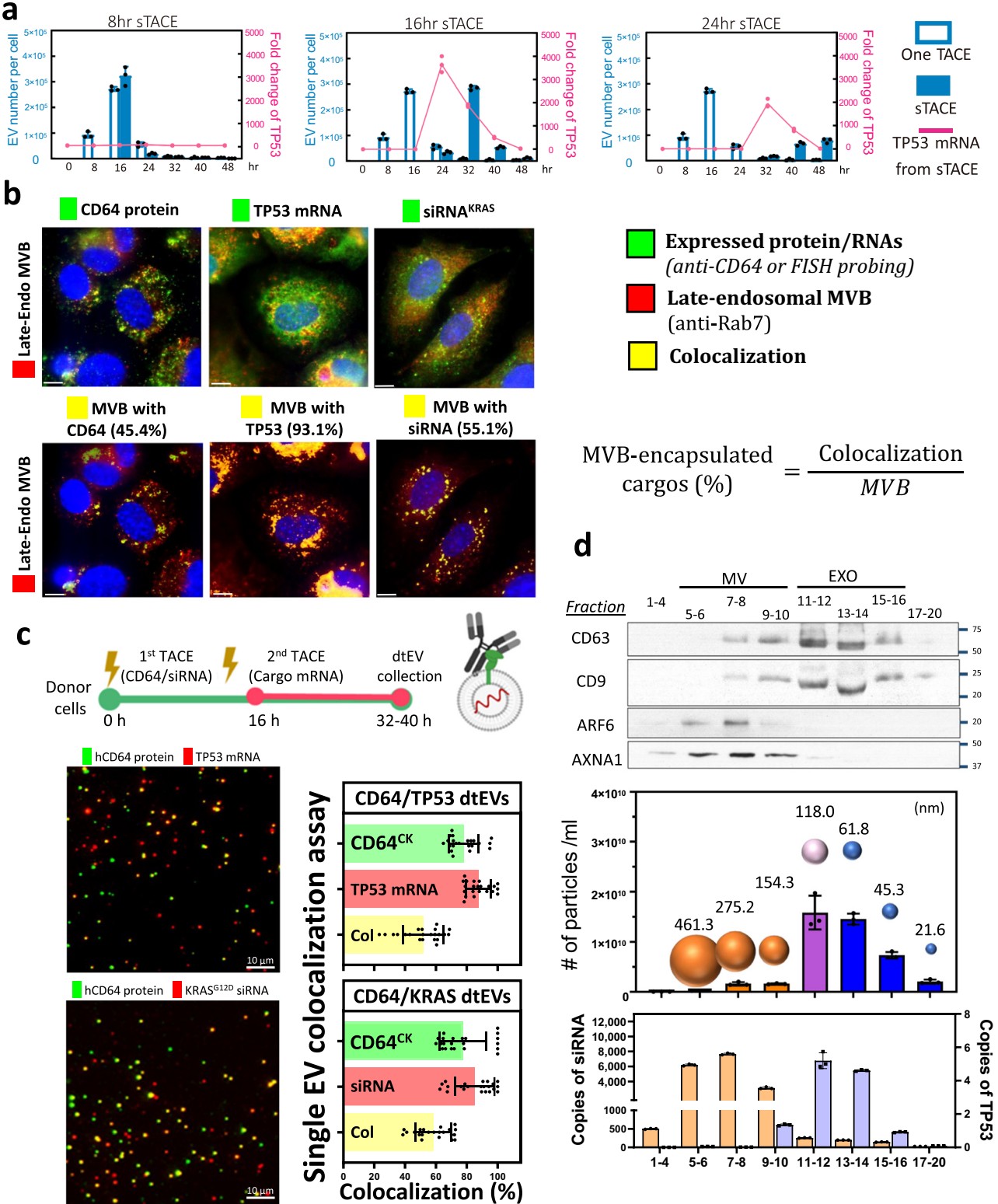

$$\text{MVB-encapsulated cargos (\%)} = \frac{\text{Colocalization}}{MVB}$$

the anti-GFP fluobody from 2 to 4 and 8 h, indicating the potential release of dtEVs lumens (Fig. 4e).

Adeno-associated virus (AAV) vectors are the highly effective gene delivery carriers owing to their strong receptor-mediated clathrin-[31] and caveolae-mediated endocytosis[32], transcytosis[33], and endosomal escape[34]. Based on our experimental results, it is tentatively suggested that the presence of specific antibody and THP targeting on the dtEV surface may enhance receptor-mediated transcytosis, endocytosis, and endosomal escape, thus resembling the behavior of viral particles.

## dtEVs carrying siKRAS$^{G12D}$ or TP53 mRNA inhibit pancreatic cancer cells

PANC-1 cells carrying KRAS$^{G12D}$ and TP53 mutations (p.R273H, loss of function) display strong resistance to first-line treatment of Gemcitabine (GEM). Treating PANC-1 cells with EVs carrying siKRAS$^{G12D}$ at $1 \times 10^6$ EVs/cell (~1 × 10$^9$ copies/cell or ~16 pg siKRAS$^{G12D}$ mass/cell) for 24 h silenced KRAS$^{G12D}$ expression at both the mRNA and protein levels, with the targeting formulation of CD64$^{ck}$_αROR1 showing the best suppression (Fig. 5a and Supplementary Fig. 15). The siKRAS$^{G12D}$

**Fig. 2 | Sequential TACE (sTACE) for dtEV production and its RNA cargo loading. a** The sTACE was performed by delivering the CD64 plasmid first and then the TP53 plasmid DNA with a time gap of 8 h, 16 h, or 24 h in MEF cells. The optimized sTACE with 16 h gap gave the best coordination of surface CD64$^{ck}$ protein, TP53 mRNA loading, and EV number for our the dtEV formulation ($n = 3$ biological independent experiments). **b** Expressed CD64 protein was stained by florescence-labeled anti-CD64 antibodies, and RNA cargoes were recognized by FISH (fluorescence in situ hybridization) probes (green). The late-endosomal MVBs were stained by florescence-labeled anti-Rab7 (red). Through the proportion of colocalized versus total fluorescent signal, we estimated that ~45% MVBs contained CD64$^{ck}$ protein and ~93% MVBs contained TP53 mRNA at 4 h, while ~55% MVBs contained siKRAS$^{G12D}$ at 12 h after TACE. Immunofluorescent images are representative of $n = 3$ biologically independent experiments. **c** TIRF fluorescence imaging of single dtEV shows ~55% of dtEVs co-expressed surface CD64$^{ck}$ and TP53 mRNA or siKRAS$^{G12D}$ cargoes when CD64$^{ck}$ plasmid was delivered 16 h apart from TP53 plasmid or simultaneously with siKRAS plasmid. The colocalization ratio is $58.5 \pm 2.4\%$ for siKRAS and $51.6 \pm 2.8\%$ for TP53 mRNA ($n = 3$ biological independent experiments, 7 or 8 images each). **d** Harvested EVs from TACE stimulated MEFs were sorted by size exclusion chromatography (qEV columns, IZON) into 20 fractions in which the EV number, averaged EV size, surface protein markers, and loaded RNA/protein cargo in each fraction were characterized. By real-time PCR analysis for RNA cargoes from each fraction, TP53 mRNA was located mainly in exosomal fractions (11-14; high CD63/CD9 expression), while siKRAS$^{G12D}$ was found in both exosomes and microvesicles ($n = 3$ biological independent experiments). Scale bars in **b** and **c** are 10 μm. Data were presented as mean ± SD. Source data are provided as a Source data file.

carrying EVs slowed down PANC-1 proliferation mainly via cell cycle arrest in the G1 phase (Fig. 5b and Supplementary Fig. 16). The combinative dtEVs containing 50/50 siKRAS$^{G12D}$ and TP53 mRNA resulted in the highest apoptotic population (Fig. 5b). The EV dose was also $1 \times 10^6$ total EVs/cell containing ~$5 \times 10^8$ copies siKRAS$^{G12D}$ and ~$4 \times 10^6$ copies of TP53 mRNA/cell or ~8 pg siKRAS$^{G12D}$ mass and ~3 pg TP53 mRNA mass/cell. The wild-type TP53 mRNA delivered by CD64$^{ck}$_ αROR1 dtEVs enhanced the expression of p53 and p21 in the recipient PANC-1 cells (Fig. 5c). Although PANC-1 cells show strong GEM resistance, the expression of wild-type p53 protein in the presence of GEM decreased anti-apoptotic BCL-xl and increased p21 expressions (Fig. 5d). The combined tEV-gene therapy and Gem-chemotherapy effectively killed PANC-1 cells, with the CD64$^{ck}$_αROR1 dtEVs containing 50/50 siKRAS$^{G12D}$ and TP53 mRNA (Fig. 5e) showing the best performance (>65% cancer cell killing efficacy under 24 h treatment) (Fig. 5f and Supplementary Fig. 17). Treatment of PANC-1 cells with dtEVs, both with and without GEM, resulted in an elevation of oncogene-induced senescence, as evidenced by an increase in SA-β-galactosidase activity (Fig. 5g and Supplementary Fig. 18). Although both mRNA and siRNA cargoes could be loaded into the same EVs by co-delivering multiple plasmid DNAs to donor cells via TACE, we prepared dtEVs with either mRNA or siRNA cargo in this study because their release profiles were different (Fig. 1c).

### dtEVs suppress orthotopic PANC-1 mice with high tumor burden

We assessed the clinical potential of dtEVs in PANC-1 tumors orthotopically implanted in the pancreas in NOD/SCID mice expressing green fluorescent protein (GFP) and luciferase. Each targeting formulation of dtEVs was fluorescently labeled (PKH26) and intraperitoneally (i.p.) injected into mice. After 24 h, major organs were collected and analyzed (Fig. 6a). Compared to LNPs and other targeted and non-targeted EVs, significantly more CD64$^{ck}$_αROR1 dtEVs accumulated in the pancreatic tumor (Fig. 6b). Fluorescence imaging (Fig. 6c) and immunohistochemical staining (Fig. 6d) of tumor sections confirm the preferential accumulation of CD64$^{ck}$_αROR1 dtEVs in the tumor over the liver and spleen.

Over 70% of pancreatic cancer patients were diagnosed at advanced stages with unresectable tumor burden[35]. We then used NOD/SCID mice for examining the therapeutic efficacy of dtEVs on high tumor burden in vivo. Animals were injected with $5 \times 10^5$ luciferase transduced PANC-1 cells per mouse and commenced treatment on Day 25. EV-based gene therapy mainly regulates cancer cell cycle arrest but not directly killing, we administered a low-dose GEM (20 mg/kg per week) to the animals. Controls include animals treated with dtEVs carrying scramble RNA (SCR), GEM alone, and LNPs containing 10-fold more synthetic TP53 mRNA and siKRAS$^{G12D}$ than those delivered by CD64$^{ck}$_αROR1 dtEVs. GEM was delivered once a week, while the rest formulations were delivered three times per week with i.p. injection of $1 \times 10^{11}$ dtEVs per dose. The estimated RNA cargo was ~$1 \times 10^{14}$ copies (~1.6 μg mass) of siKRAS$^{G12D}$ or scramble RNA. Of all treatments, GEM together with CD64$^{ck}$_αROR1 dtEVs carrying TP53 mRNA and

siKRAS$^{G12D}$ strongly suppressed tumor growth and prolonging the overall survival of animals (Fig. 6e–g). For the combinative siKRAS$^{G12D}$ and TP53 mRNA formulation, $5 \times 10^{10}$ dtEVs carrying either siKRAS$^{G12D}$ or TP53 mRNA were mixed before use. The estimated RNA cargo was ~$2 \times 10^{11}$ copies (~0.3 μg mass) of TP53 mRNA and ~$5 \times 10^{13}$ copies (~0.8 μg mass) of siKRAS$^{G12D}$. Immunohistochemical staining of residual tumor tissue revealed that dtEV treatment restored TP53 expression and silenced KRAS expression (Fig. 6h and Supplementary Fig. 19). The dtEVs were biocompatible and displayed good blood circulation profiles with minimal cellular toxicity (Supplementary Fig. 20a, b). Unlike GEM or LNP treatment, the EV-treated mice showed much less weight loss (Supplementary Fig. 20c).

### Targeting and therapeutic efficacy of dtEVs in subcutaneous PDX mouse models

We then tested the targeting and therapeutic performance of dtEVs in patient-derived xenograft (PDX) PDAC tumors inoculated subcutaneously (subQ) into the back of NOD/SCID mice. We selected a subQ mouse model first, because the animal viability and quality in the orthotopic PDX NOD/SCID mice was very poor due to the major surgery in animal model preparation. One subQ animal was intratumorally injected with $1 \times 10^{11}$ fluorescent-labeled IgG_EVs and CD64$^{ck}$_αROR1 dtEVs. Consistent with the transcytosis and spheroid penetration results in Fig. 3g, dtEVs (CD64$^{ck}$_αROR1) spread more broadly throughout the tumor than IgG_EVs (Fig. 7a). Biodistribution in subQ mice via i.p. injection also showed stronger dtEV accumulation in tumor comparing to non-targeted IgG_EVs and non-targeted LNPs (Fig. 7b, c). Encouraged by these results, we further investigated the dtEVs properties in a genotype matching PDX model (ROR1$^{++}$, KRAS$^{G12D}$, and TP53$^{A138V}$ mutations). We subcutaneously inoculated a $3 \times 3 \times 3$ mm$^3$ PDX tissue into the back of NOD/SCID mouse. By three-week treatment, dtEVs containing TP53-mRNA and siKRAS$^{G12D}$ (dtEVs: $1 \times 10^{11}$/100 μl, i.p. injection, three-time/week with ~$2 \times 10^{11}$ copies or ~0.3 μg mass of TP53 mRNA and ~$5 \times 10^{13}$ copies or ~0.8 μg mass of siKRAS$^{G12D}$ per injection) greatly inhibited the large tumor growth, while combining a low dose of GEM (20 mg/kg, once/week) could further suppress the tumor activity (Fig. 7d,e). KRAS knockdown using Western blot on the collected tumor tissues was observed (Supplementary Fig. 21). Hematoxylin and eosin staining of various tissues obtained from the treated animals revealed cancer metastasis in the lung, pancreas, and fat tissue (Supplementary Fig. 22). However, animals treated with dtEVs containing TP53 mRNA and siKRAS$^{G12D}$ had substantially fewer and smaller metastatic lesions, particularly with GEM (Fig. 7f, g and Supplementary Fig. 23). It is well-established that aberrant activation of mutant KRAS can override oncogene-induced senescence (OIS), thereby promoting the survival and proliferation of cancer cells[36,37]. In the tumor tissues and metastatic lung lesions of mice treated with dtEV or dtEV+GEM, we observed an increase in β-galactosidase activity and a decrease in K-i67 expression (Fig. 7h, i). These findings indicate a reactivation of OIS by dtEVs to suppress tumor growth.

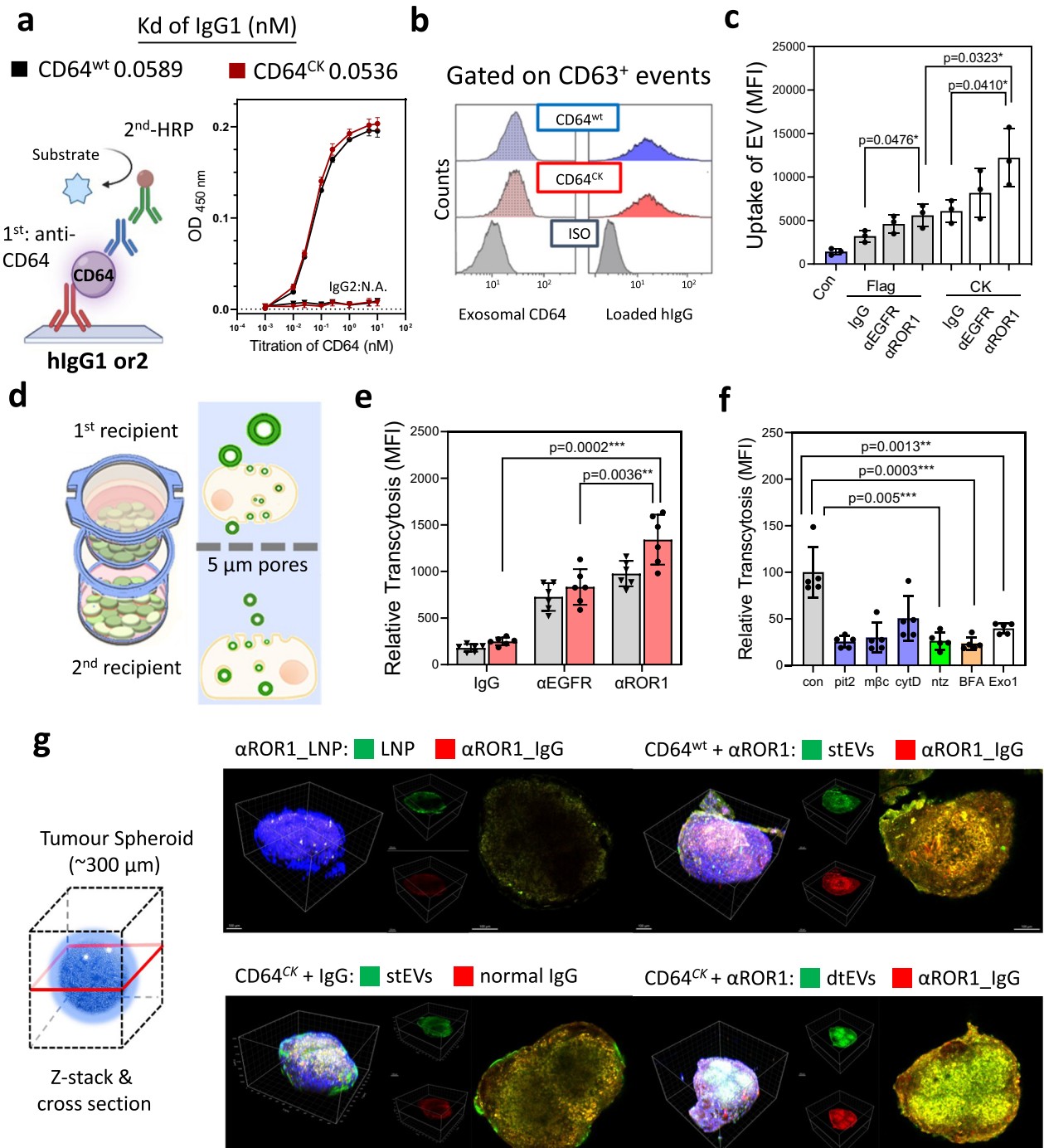

**Fig. 3 | Penetration and uptake characteristics of dtEVs in PANC-1 cells.**
**a** Enzyme-linked immunoassay (ELISA) shows both CD64$^{wt}$ ($K_d$ = 0.0589 nM) and CD64$^{ck}$ ($K_d$ = 0.0536 nM) bound to hIgG1, but not hIgG2, with high affinity ($n$ = 3 biological independent experiments). **b** Flow cytometry analysis using anti-CD63 beads to capture dtEVs confirms surface CD64$^{ck}$ bound to hIgG. **c** Internalization assay using EVs labeled with fluorescent PKH67 dye shows higher cell uptake of dtEVs loaded with humanized anti-ROR1 mAb (αROR1) than humanized anti-EGFR (αEGFR) or control IgG. Uptake of dtEVs with CK peptides (12,209 ± 1914) doubled that of Flag control (5,585 ± 755.9) ($n$ = 3 biological independent experiments). **d** Schematic of transcytosis assay used to quantify the entry and exit of various nanocarriers from the top to the bottom cell monolayer separated by a Transwell® insert membrane with 5 μm pores. **e** CD64$^{ck}$_ αROR1 dtEVs displayed the greatest putative transcytosis ($n$ = 5 biological independent experiments). **f** Pre-treating top

cells with inhibitors of clathrin-mediated endocytosis (Pitstop 2, Pit2, 10 mM), caveolae-mediated endocytosis (Methyl-β-cyclodextrin, mβC, 10 mM) or exosomal secretion (neticonazole, Ntz, 10 mM) greatly prevented this putative transcytosis, but not inhibitor of macropinocytosis (Cytochalasin D, cytD, 10 μM). Con: no inhibitor ($n$ = 5 biological independent experiments). **g** dtEVs penetrated tumor spheroids of PANC-1 cells (~300 μm) deeper than stEVs or LNPs after 24 h incubation (blue: DAPI). LNPs and EVs were labeled with PKH67 (green), while anti-hIgG was in red. Immunofluorescent images are representative of $n$ = 3 biologically independent experiments. Scale bar in **g**: 100 μm. Data were presented as mean ± SD. For **c–f**, the data were analyzed by unpaired two-sided Student's $t$ test. (*$p$ < 0.05, **$p$ < 0.01, ***$p$ < 0.001, ****$p$ < 0.0001). Source data are provided as a Source data file.

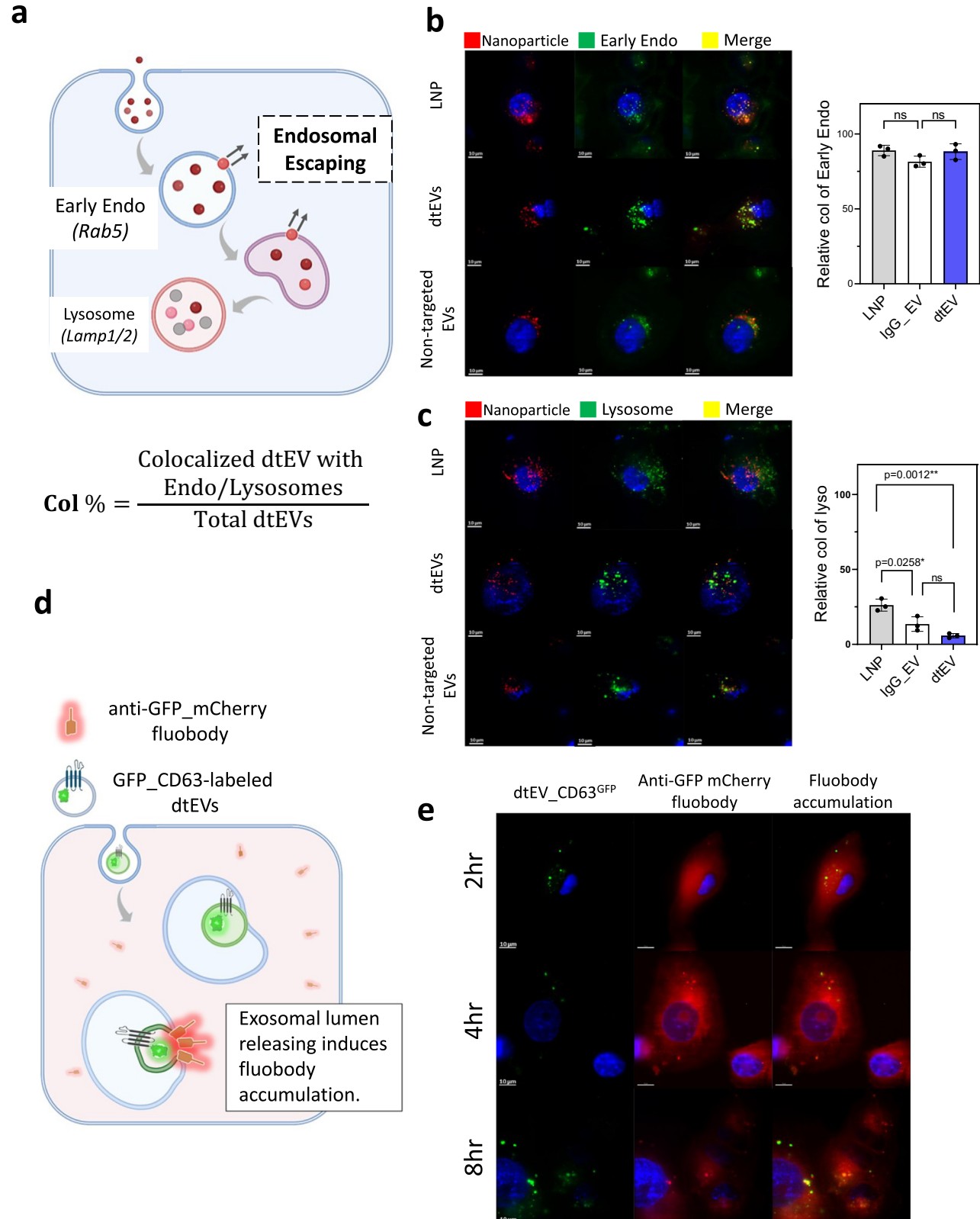

$$\text{Col \%} = \frac{\text{Colocalized dtEV with Endo/Lysosomes}}{\text{Total dtEVs}}$$

## Therapeutic efficacy of dtEVs in orthoptic PDX mouse models

Because the orthotopic and subcutaneous PDAC pathology is quite different, especially the physiological barriers, we further prepared an orthotopic tumor by stitching a 3 x 3 x 3 mm³ PDX tissue from a different PDAC patient to the pancreatic tail of BALB/c-Nude mice which are much more robust than NOD/SCID mice in the animal model preparation. We also increased the suggested GEM dose from 20 to 50 mg/kg per week because of the different mouse types. The genotyping of PDX tissue carried both KRAS[G12D] and TP53[R196*] gene mutations. TP53[R196*] mutation has worse prognosis due to the loss of the entire cancer suppression function[38]. Furthermore, this PDX tissue also expressed a high level of the Glioma-associated homologue-1 (Gli1) protein known to regulate the Hedgehog pathway. Overexpression of Gli1 accelerates KRAS-initiated pancreatic tumourigenesis[39,40].

**Fig. 4 | Trafficking of dtEVs in recipient cells. a** Nanocarriers labeled with PKH26 (red) were taken up by endocytosis and underwent vesicular trafficking in the recipient cells, from the endosome to the lysosome. **b** PANC-1 cells were initially treated with fluorescent nanoparticles (red) for 2 h and then fixed for co-staining with anti-Rab5 to visualize the colocalization in early endosome (green). Colocalization in recipient cells showed a similar level for LNPs, non-targeted IgG-EVs and dtEVs (colocalization % (Col%): LNP: 89.0 ± 3.5; non-targeted IgG-EV: 88.17 ± 5.2; dtEV: 81.50 ± 3.7, n = 3 biological independent experiments). **c** When comparing the colocalization (yellow) of LNPs, dtEVs, and non-targeted IgG-EVs, we observed less colocalization of dtEVs in lysosomes (green) after 8 h incubation (Col%: LNP: 26.17 ± 4.0; non-targeted IgG-EV: 13.53 ± 4.87; dtEV: 5.73 ± 1.45, n = 3 biological independent experiments). **d** A fluobody recognition system was used to monitor the interaction of dtEV lumens within vesicular trafficking. Recipient cells were pre-transfected with an anti-GFP fluobody tagged with mCherry (red), which is expressed in the cytosol. Upon release of the dtEV lumen, the cytosolic anti-GFP fluobody could recognize CD63$^{GFP}$ on the dtEV lumens, leading to temporary local accumulation. **e** PANC-1 cells expressing anti-GFP_mCherry fluobody were incubated with dtEVs containing CD63$^{GFP}$ for 2, 4, and 8 h. The induced accumulation of anti-GFP fluobody was observed to increase from 2 to 4 and 8 h, suggesting the potential of lumen releasing of dtEV. Fluorescent images are representative of n = 3 biologically independent experiments. Scale bar in **b**, **c**, and **e** are 10 μm. Data were presented as mean ± SD. For **b** and **c**, the data were analyzed by unpaired two-sided Student's t test. (ns not significant, *p < 0.05, **p < 0.01, ***p < 0.001, ****p < 0.0001). Source data are provided as a Source data file.

Suppressing Gli1 activity leads to oncogenic inhibition and enhanced Gemcitabine-induced cytotoxicity in PDAC cell lines[41,42] (Supplementary Fig. 24). We intraperitoneally injected dtEVs to the animals 14 days after orthotopic tumor tissue implantation. Recipient mice were treated with 50 mg/kg GEM once per week and 1 x 10$^{11}$ dtEVs/100 μl three times per week containing either 50/50 TP53-mRNA/siKRAS$^{G12D}$ (dtEV) or 1/1/1 TP53-mRNA/siKRAS$^{G12D}$/siGli1 (dtEVs$^{Gli1}$) and monitored by positron emission tomography (PET) (Fig. 8a). Comparing to the saline control group, dtEVs carrying TP53-mRNA and siKRAS$^{G12D}$ suppressed the orthotopic tumor burden after two-week treatment, and adding siGli1 in dtEVs$^{Gli1}$ could further slowdown the orthotopic tumor growth (Fig. 8b and Supplementary Fig. 25). We also conducted H&E and IHC staining for Ki-67, pERK, p21, KRAS$^{G12D}$, and Gli1 in orthotopic tumor tissues from each treatment group. Both dtEV and dtEVGli1 with GEM suppressed the expression Ki-67, pERK, p21, and KRAS$^{G12D}$. Treatment of dtEVGli1+ GEM further decreased Gli1 expression (Fig. 8c). Together, our results show that dtEVs carrying multiple gene cargoes together with chemotherapy can effectively suppress large PDAC tumors with complicated genetic profiles.

## Discussion

Current EV therapy faces challenges in production, targeting, and cargo encapsulation. Although therapeutic siRNAs can be loaded into pre-isolated EVs by passive diffusion or external forces such as electroporation[11,43], this approach is inefficient for large mRNAs. Genetically modified cells for cooperative desired mRNA in secreted EVs[14] is associated with inefficiencies due to low rates of genetic modification and concerns regarding the introduction of xenogenic RNAs and proteins. Additionally, both approaches face challenges in terms of low EV secretion and limited RNA/protein loading, hampering their clinical applicability. In this study, we propose a clinically accessible, scalable, and cost-effective method for generating a large quantity of EVs containing CD64CK protein through asymmetric cell electroporation using lab consumable Transwell® inserts (TACE). We demonstrate that transfecting selected cells with plasmid DNAs enables high rates of EV secretion, carrying both transcribed mRNA/siRNA and translated proteins of interest. This is achieved through the rapid non-endocytic delivery of plasmid DNA molecules via nanopores at high electric field strengths, which preserves cell viability and facilitates abundant release of EVs while maintaining the integrity of the cell membrane. Moreover, we achieve high loading of transcribed therapeutic mRNA (such as TP53) or siRNA (such as siKRAS$^{G12D}$, siGli1), as well as translated membrane protein (such as CD64), in the secreted EVs. This is possible because the normal endo-lysosomal and autophagy pathways in transfected cells are unable to immediately degrade the unusually large quantity of expressed biomolecules.

We achieved dual targeting on the EV surface through an engineered CD64 protein that has both tissue-homing peptides (THP) and humanized monoclonal antibodies (hmAb) with high biocompatibility. While post-insertion of polyethylene glycol (PEG)-grafted targeting molecules can be employed with EVs[44], achieving high loading as shown in our dtEVs is challenging. We also showed this double targeting approach significantly improved tumor targeting, tumor tissue penetration, and cellular uptake. Our dtEVs carrying multiple RNA cargoes, offer a superior delivery system in gene delivery.

Liposomal formulations of anticancer agents are successful nanocarriers in cancer therapy by prolonging the circulating lifetime of the loaded cargo and enhancing deposition within tumor[45]. Most liposomes achieve endosomal escape in tumor cells by destabilizing endosomal compartments, resulting in efficient cargo release primarily in the outer layer of the tumor tissue. Recent studies have demonstrated that EVs can remain stable within endosomes and release cargo upon interaction with the endosomal membrane[29]. In Fig. 4, we also observed a similar phenomenon during intracellular trafficking of dtEVs. Additionally, we observed membrane fusion, implying lumen release of dtEVs within endosomes. Our experimental results support the superior performance of dtEVs in terms of tumor uptake, penetration, and inducing strong cancer cell cycle arrest. However, further investigation is necessary to gain a better understanding underlying endosome escape and transcytosis of dtEVs, as well as to quantify their advantages in the treatment of solid tumors.

In our PANC-1 and PDX mouse models, we showed that together with a first-line chemotherapy drug, Gemcitabine, our dtEVs could substantially inhibit large solid tumors and metastatic lesions. However, we still observed the tumor regrowth after 49 days of treatment in the PANC-1 orthotopic mouse model (Fig. 6). Similarly, a combination of the best dtEV formulation (1/1/1 TP53-mRNA/siKRAS$^{G12D}$/siGli1 and a higher dose of Gemcitabine (50 mg/kg) did not fully suppress the tumor growth in an aggressive orthotopic PDX PDAC mouse model (Fig. 8). These findings suggest that solely rescuing TP53 and suppressing a few oncogenes at a fixed drug dosage for an extended period of time is insufficient to achieve complete sensitization of PDAC to standard chemotherapy. Further investigation is required to delve into detailed mechanisms of gene mutations in cancer cells and their correlation to chemodrug resistance. This will enable the design of an optimal dtEV-gene formulation and chemodrug combination, tailored to individual cancer patients, with appropriate dosages. Given the heterogeneity of cancer cells within tumors and the potential for more mutations to arise during treatment, timely adjustments to the dtEV-gene formulation and dosage are necessary. In addition to conventional chemotherapy drugs, our EV-based gene therapy holds the potential to be combined with other cancer treatment modalities.

Additionally, targeted EVs generated through TACE can be utilized to deliver specific coding and non-coding genes from diverse cell sources, offering potential applications in various diseases such as neurodegenerative disorders, autoimmune conditions, infectious diseases, and rare disorders.

## Methods

### Cell culture and in vitro treatment

Mouse embryonic fibroblasts (MEFs) and PANC-1 cell line were from Millipore and American Type Culture Collection (ATCC, USA) respectively. Cells were cultured in Dulbecco's Modified Eagle's Medium (DMEM) (Thermo Fisher, Scientific) containing 10% heat-inactivated

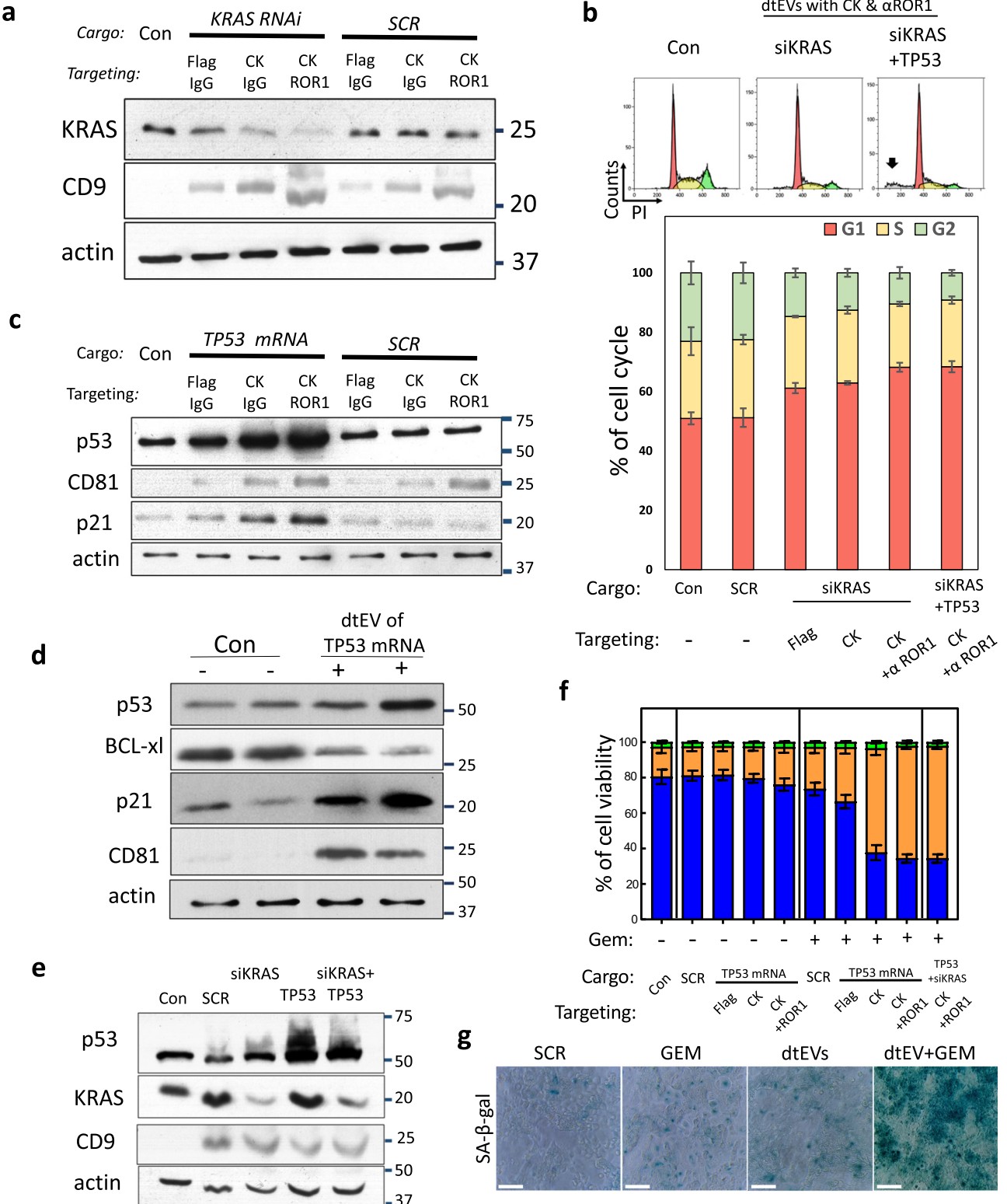

Fetal Bovine Serum (FBS) (Thermo Fisher Scientific). Human bone marrow-derived mesenchymal stem cells (hBMSCs) were purchased from RoosterBio (Frederick, MD) and were cultured in cell culture media and supplements (RoosterNourish™ MSC) with a mixture of 1% penicillin and streptomycin. All cell cultures were at 37 °C in humidified conditions equilibrated with 5% $CO_2$. The inhibitors for heat shock proteins and ROS were purchased from Sigma-Aldrich and used to pretreat donor cells for 6 h before TACE.

**Plasmids construction**
Plasmids expressing CD64[WT], TP53[WT], shKRAS[G12D], scramble, and GFP were constructed by VectorBuilder Inc (Chicago, IL). Tissue homing peptides were cloned into the N-terminus of CD64 coding sequence in the expressing plasmid as construction map. All plasmids were extracted using plasmid kits (QIAGEN) and their concentration and quality were confirmed by NanoDrop™ 2000 (Thermo Fisher Scientific).

**Fig. 5 | In vitro efficacy of dtEVs carrying siKRASG12D or TP53 mRNA. a** Western blot of PANC-1 cells treated with various dtEVs carrying endogenous siKRASG12D or scramble siRNA (SCR) for 24 h shows silenced KRASG12D expression. **b** Flow cytometry histogram of DNA distribution in PANC-1 cells treated with CD64ck_αROR1 dtEVs carrying siKRASG12D or 50/50 siKRASG12D/TP53 mRNA. Significant apoptosis (black arrow) is seen with cells treated with dtEVs carrying two cargoes. Lower panel shows percentage of cells at G1, S and G2/M stages of the cell cycle in PANC-1 cells treated with various stEVs (Flag, CK) and dtEVs (CK + αROR1) carrying either siKRASG12D alone or 50/50 siKRASG12D/TP53 mRNA. **c** Western blot of PANC-1 cells treated with various dtEVs carrying either endogenous TP53 or scramble (SCR) mRNA for 24 h. The p53 and p21 expressions were upregulated in cells treated with dtEVs carrying TP53 mRNA. **d** Western blot of PANC-1 cells treated with Gemcitabine (GEM) and CD64ck_αROR1 dtEVs carrying endogenous TP53 mRNA for 24 h

shows upregulated p53 and p21, and downregulated BCL-xl expressions. **e** Western blot of PANC-1 cells treated with GEM and CD64ck_αROR1 dtEVs carrying SCR, siKRASG12D, TP53 mRNA, or 50/50 siKRASG12D/TP53 mRNA for 24 h. **f** Percentage of live, dead and apoptotic (Apop) PANC-1 cells treated with various stEVs (Flag, CK) and dtEVs carrying TP53 mRNA in the presence (+) or absence (−) of GEM. Cells treated with CD64ck_αROR1 dtEVs and GEM led to high cancer cell death. **g** The treatment of PANC-1 cells with dtEVs, both with and without GEM, elevated oncogene-induced senescence, as evidenced by increased in SA-β-galactosidase activities. Con is EVs from untreated native MEFs. All EVs were delivered at $1 \times 10^6$ EVs/cell. All error bars represent s.e.m. over three independent samples. The siRNA sequences are given in Supplementary Table 3. Scale bar in **g** is 100 μm. **a, c, d, e, g** Western blots and staining images are representative of three biological independent experiments.

## Transwell®-based asymmetric cell electroporation and EV collection and characterization

MEFs or hBMSCs were cultured on the membrane surface of a Transwell® insert overnight. Plasmids were preloaded in the bottom chamber and then delivered into cells via membrane pores under an electric field by Xcell System (Bio-rad). After TACE, transfected donor cells were cultured on the Transwell® insert in the serum-free medium for 24 h for EV collection. The harvested media was centrifuged (2000×*g*, 30 min) to remove cell debris, and a tangential flow filtration (TFF) system (D02-S500-05-S, Repligen) was used to enrich and diafiltrate EVs under a sterile condition to remove soluble proteins, lipid proteins, and small vesicles. The EV-enriched medium was further condensed by Amicon® Centrifugals (Merck Millipore) with 3 kDa MWCO at 3,000× g to a desirable volume and EV concentration. The collected cell culture medium after TACE went through DNase treatment to remove the possible plasmid DNA contamination. We also designed a serial qPCR assay to measure the possible plasmid DNA contamination (Supplementary Fig. 26). The plasmid DNA was measured by three different qPCR primer sets as (i) Exon 10-11 of hTP53 (Hs01034249_m1, ThermoFisher), (ii) AmpR (Mr00661613_cn, Thermofisher), and (iii) structural sequence of plasmid DNA (blue). The qPCR measurement of Exon 10-11 of hTP53 can detect both plasmid DNA and mRNA of TP53 in the sample, while the detection region of AmpR and Structure Seq only exits in plasmid DNA.

For qRT-PCR or Western blotting assays, EVs were packed as pellets from culture supernatants by incubating with a total exosome isolation reagent (Invitrogen) overnight and followed by centrifugation at 10,000×*g* for 60 min at 4 °C. The loading amount of RNA was comparable when using either TFF or TEI purification (Supplementary Fig. 27). The microvesicle and exosome subpopulations were sorted by size exclusion chromatography (qEV Size Exclusion Column, Izon Science) according to manufacturer's suggestion. The EV size and particle numbers were measured by dynamic light scattering goniometry (DLS, BI-200SM Goniometer, Brookhaven Instruments.) and NanoSight (NTA, Malvern Panalytical Ltd, UK). Measurements of dtEVs by DLS and NTA were compared and are given in Supplementary Fig. 28. For antibody-targeted EVs, the EVs were incubated with a selected humanized monoclonal antibody (hmAb) overnight at 4 °C. Free hmAbs were then removed by ultracentrifugation. The RNA loading in EV needs to follow the RNase with or without Triton X-100 treatment procedure to be validated (Supplementary Fig. 29). The use of different EV purification and characterization methods would affect the measured EV numbers and RNA loading level, but the quantities are in the similar range.

## Lipoplex nanoparticle preparation

The synthetic mRNA or siRNA was encapsulated in PEGylated liposomes (Cat#: PEGLIV, Altogen Biosystem). The 100 μl diluted RNA (1 μg/μl RNase-free water) was mixed with 50 μl Transfection reagent for 15 min and then with 10 μl of Transfection Enhancer Reagent for 5 min at room temperature (RT). Glucose (5%, w/v) isotonic solution

was used to finalize the injection volume to 0.4 ml per mouse. The mRNA and siRNA doses were 40 and 1000 μg/kg/day respectively in animal studies. A post-insertion method was adopted to incorporate anti-ROR1 PEGylated antibody onto LNPs[46].

## RNA extraction and analysis

RNA from cells or EV pellets was extracted by total RNA purification kits (Norgen Biotek) and RNase-free DNase I (Cat#: 18047019, Thermo Fisher Scientific) to remove plasmid residues. Complementary DNA (cDNA) was generated by a high-capacity cDNA reverse transcription kit (Cat#: 4368814, Thermo Fisher Scientific) from extracted RNA and measured by Real-time polymerase chain reaction with TaqMan (Thermo Fisher Scientific) expression assay for TP53 (HS01034249 and HS01034254), KRAS (Hs00364284_g1), GAPDH (Hs02786624_g1 and Mm99999915_g1), and siRNA of KRASG12D (siKRASG12D) (Custom TaqMan order) as internal control genes. A synthetic TP53 mRNA was also purchased (TriLink Biotech, San Diego, CA) as an internal control gene. The signal was measured on a ViiA 7 Real-Time PCR System (Thermo Fisher Scientific). The expression of EV RNA of interest relative to the internal control gene was calculated by using the threshold cycle number (Ct). The relative EV gene expression for each condition was normalized to vehicle control and fold change determined by using the comparative method (2ΔΔCt). For mRNA integrity assay, EV mRNA was first quantified by Qubit RNA HS Assay Kit (Q32852, Thermo Fisher Scientific) with Qubit 4 Fluorometer (Thermo Fisher Scientific) and diluted to 1000 μg for Bioanalyzer loading. The EV mRNA integrity and size distribution were analyzed using an Agilent 2100 Bioanalyzer with an RNA 6000 Pico kit (Agilent Technologies).

## EV RNA profiling

EV microRNA was isolated using the miRNeasy kit (QIAGEN, Germantown, MD) and the concentrations of RNA was assessed on Bioanalyzer with an RNA (Pico) chip (Agilent Technologies, Santa Clara, CA). An in-house small RNA sequencing library construction method was used to characterize the microRNA in samples (PMID: 29388143). Proper insert size of the library was selected with Pippin HT (Sage Science, Beverly, MA) and the concentration was measured using NEBNext Library Quant Kit (New England Biolabs, Ipswich, MA). The library concentrations were adjusted and pooled to a final concentration of 2 nM then run on a NEXTseq DNA sequencer (Illumina, San Diego CA). The small RNA (sRNA-Seq) data was analyzed with sRNAnalyzer (PMC5716150). The quantity of individual microRNAs was determined based on the number of mapped reads that were adjusted with Count Per Mapped Million (CPM). EV mRNA was characterized using Agilent 8x60 microarray and fluorescent probes were prepared from isolated RNAs using Agilent QuickAmp Labeling Kit according to the manufacturer's instructions (Santa Clara, CA). Gene expression information was obtained using Agilent's Feature Extraction and processed with the Institute for Systems Biology's in-house SLIMarray pipeline (PMC1636632).

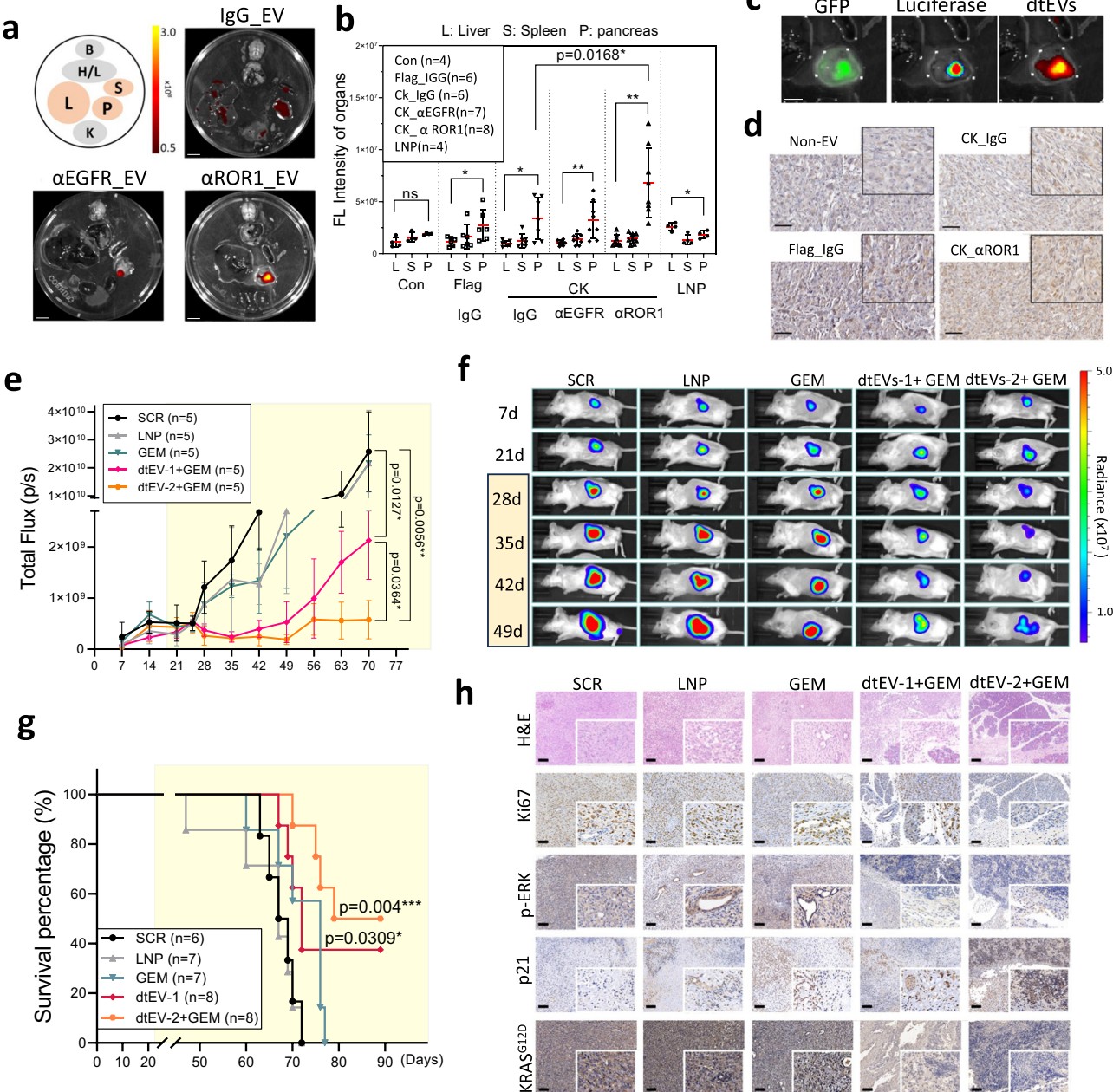

**Fig. 6 | Targeting and therapeutic efficacy of dtEVs in mice bearing orthotopic PANC-1 tumors. a** Organ imaging shows preferential accumulation of PKH26-labeled dtEVs in orthotopic PANC-1 tumors in NOD/SCID mice [B: brain; H/L: heart and lungs; L: liver; S: spleen; P: pancreas; K: kidneys, scale bar: 10 mm]. **b** Tissue distribution analysis reveals PKH26-labeled dtEVs accumulate preferentially in the pancreas, with the highest accumulation seen using CD64$^{ck}$_αROR1 dtEV. **c** Co-localization of fluorescence signal from luciferase/GFP transfected PANC-1 and PKH26-labeled EVs confirms high dtEV accumulation in the orthotopic PANC-1 tumor (scale bar: 5 mm). **d** Immunohistochemical staining of residual PANC-1 tumor tissue treated with different EVs shows a strong accumulation of dtEVs (brown: anti-hCD64, scale bar: 50 μm). **e, f** Total flux of In Vivo Imaging System (IVIS) (**e**) and whole-animal imaging (**f**) over time shows combined GEM and dtEVs carrying both siKRAS$^{G12D}$ and TP53 mRNA treatment was best at suppressing the growth of advanced tumors. The NOD/SCID mice (n = 5 mice each group) were i.p. injected with $1 \times 10^{11}$ dtEVs carrying either

scramble RNA, siKRAS$^{G12D}$ (dtEVs-1), 50/50 siKRAS$^{G12D}$/TP53 mRNA (dtEVs-2) combined with GEM (15 mg/kg), or lipid nanoparticles (LNPs) loaded with 10-fold more synthetic TP53 mRNA and siKRAS$^{G12D}$. Both dtEVs and LNPs were injected 3 times per week, while GEM was injected once a week. One other control cohort was i.p. injected 15 mg/kg Gemcitabine alone (GEM) once per week. **g** dtEVs significantly extended the survival of mice bearing PANC-1 orthotopic tumors (n of SCR, LNP, GEM, dtEV-1, and dtEV-2 + GEM are 6, 7, 7, 8, and 8 mice). *$P < 0.05$, **$P < 0.01$, log-rank test after Bonferroni correction. **h** Haematoxylin and eosin (H&E), Ki67, p-ERK, p21, KRAS$^{G12D}$ staining of treated pancreatic tumors. Yellow marked regions in **e** and **g** are the drug treatment period. Data were presented as mean ± SD. For **b** and **f**, the data were analyzed by unpaired two-sided Student's t test. (*$p < 0.05$, **$p < 0.01$, ***$p < 0.001$, ****$p < 0.0001$). Source data are provided as a Source data file. The siRNA sequences are given in Supplementary Table 3. Scale bar in **h** is 100 μm. **a, c, d, h** fluorescent and immunohistochemistry images are representative of three independent mice.

## Intracellular staining and co-localization analysis

Cells after TACE with a selected plasmid were placed on 35 mm confocal dishes and fixed in 4% formaldehyde solution. The fixed cells were permeabilized with 0.2% (v/v) Triton X-100 (Sigma-Aldrich) for 2 min. For mRNA staining, cells were incubated with blocking buffer (R37520,

Thermo Fisher Scientific) for 1 h at RT and then incubated with 1 μM FISH (Fluorescence In Situ Hybridization) probes for TP53 mRNA (Qiagen) for 1 h at 37 °C. Cells were sequentially incubated with fluorescence-labeled primary antibodies: anti-Rab7 (Cell Signaling, 94298S), anti-CD64 (MA5-16437, Thermo Fisher Scientific) in 1% BSA solution after

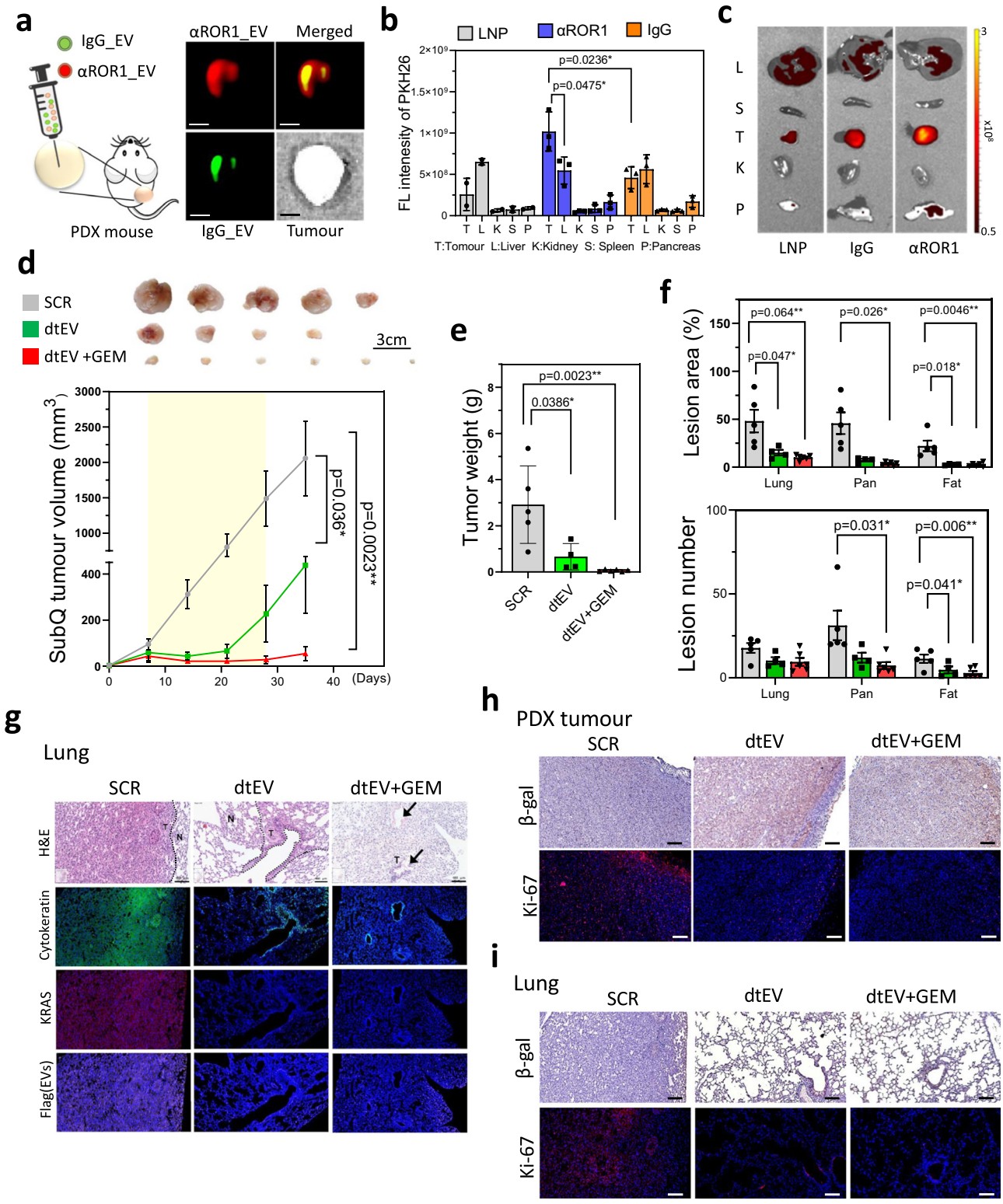

blocking with 5% BSA in PBS solution for 1 h at RT. Fluorescent images of stained cells were taken by Total Internal Reflection Fluorescence microscopy (TIRFM, N-SIM S, Nikon) and quantitated for co-localization by Imaris software (BITPLANE, Oxford Instruments, Zurich, Switzerland).

**β-galactosidase activity assay**

PANC-1 cells were seeded on 24-well plates and treated with dtEVs with or without GEM for 24 h. After the treatment, the cells were fixed and stained for β-galactosidase activity following the manufacturer's protocol (Cat#: 9860 S, Cell Signaling). The development procedure was carried out overnight at 37 °C without $CO_2$. The quantification of β-galactosidase activity was measured at $OD_{420nm}$ using the Spark® Cyto plate reader (Tecan, Switzerland).

**Immune lipoplex nanoparticle imaging assay**

RNA and protein co-localization in single EVs were measured using molecular beacons (MBs) encapsulated within cationic lipoplex nanoparticles and fluorescence-labeled antibodies to bind with antibody captured individual EVs on a biochip surface and optically detected by TIRFM. Details are given in Supplementary Fig. 3.

**Fig. 7 | Targeting and therapeutic efficacy of dtEVs in PDX mouse models.**
**a** Schematic (left) and fluorescence image (right) of subcutaneous PDX tumor intratumorally co-injected with PKH67-labeled IgG stEV (green) and PKH26-labeled CD64$^{ck}$_αROR1 dtEV (red). Broader tumor penetration is seen with dtEV. (Scale bar is 5 mm) **b** Biodistribution (left) of PKH-labeled LNP, stEV (IgG) and dtEV (αROR1) in PDX mice bearing subcutaneously implanted pancreatic tumors shows dtEV accumulated preferentially in the tumor (n of LNP, stEV_IgG and dtEV_αROR1 = 2, 3, and 3 mice). **c** Organ imaging (right) confirms the tumor accumulation. Change in **d**, volume and **e**, weight of PDX tumors (with KRAS$^{G12D}$ and TP53$^{A138V}$ mutations and EGFR$^+$/ROR1$^+$ expression) subcutaneously implanted in NOD/SCID mice after a 3-week treatment (yellow zone) with CD64$^{ck}$_αROR1 dtEVs carrying scramble RNA (SCR), CD64$^{ck}$_αROR1 dtEVs carrying 50/50 TP53 mRNA and siKRAS$^{G12D}$ (dtEV), or CD64$^{ck}$_αROR1 dtEVs carrying 50/50 TP53 mRNA and siKRAS$^{G12D}$ plus 20 mg/kg GEM once per week. EVs were delivered at a dose of $1 \times 10^{11}$ EVs 3 times per week. (n of SCR, dtEV, and dtEV+GEM = 5, 4, and 6 mice).

**f** Number and area of metastatic tumor lesions in different organs (n of SCR, dtEV, and dtEV+GEM = 5, 4, and 6 mice). **g** Lung sections from treated PDX mice with hematoxylin and eosin (H&E) stain, DAPI (blue, nuclei), anti-cytokeratin (green, tumor marker), anti-KRAS (red), and anti-Flag (purple, delivered EV marker). Metastatic lesions in the lung were decreased by dtEVs with or without GEM [N: normal; T: tumor tissue]. **h** Tumor tissues of mice treated with dtEV or dtEV+GEM exhibited increased β-galactosidase expression and decreased K-i67 expression. **i** The therapeutic effect, characterized by increased β-galactosidase expression and decreased K-i67 expression, was also observed in metastatic lesions in the lung. The treatment period is highlighted in yellow in **d**. (*P < 0.05, **P < 0.01, ****P < 0.0001, as determined by unpaired two-sided Student's t test. P-values are provided for selected comparisons.) The siRNA sequences are given in Supplementary Table 3. Source data are provided as a Source data file. (Scale bar of figures **g**, **h**, and **i**: 100 μm) **g–i** fluorescent and immunohistochemistry images are representative of three independent mice.

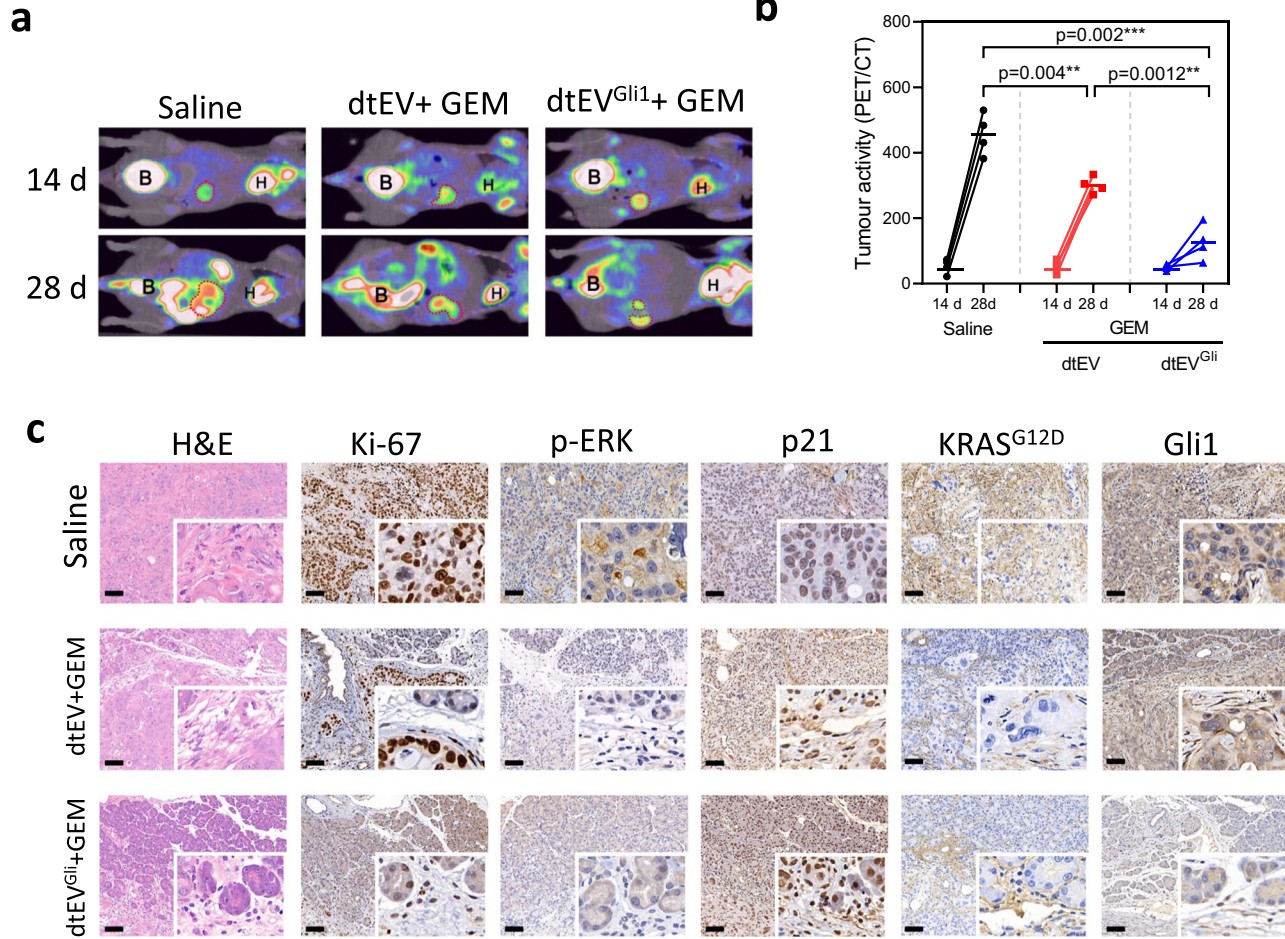

**Fig. 8 | Therapeutic efficacy of dtEVs in an orthotopic PDX mouse model.**
**a** Representative PET-CT images displaying the growth of orthotopic PDAC tumors in the abdominal cavity of mice (n = 4 mice each group). **b** Biweekly monitoring of orthotopic PDX pancreatic tumors (with KRAS$^{G12D}$ and TP53$^{R196*}$ mutations, and high ROR1 and Gli1 expressions) in Balb/C-nu mice subjected to different treatments: saline, CD64$^{ck}$_αROR1 dtEVs carrying 50/50 TP53 mRNA and siKRAS$^{G12D}$ combined with 50 mg/kg/week GEM (dtEV + GEM), or CD64$^{ck}$_αROR1 dtEVs carrying 1/1/1 TP53 mRNA/siKRAS$^{G12D}$/siGli1 combined with 50 mg/kg GEM (dtEV$^{Gli1}$ + GEM). The dtEVs were administered at a dose of $1 \times 10^{11}$ EVs three times per week, while GEM was

administered once a week. Each cohort consisted of four mice (n = 4 mice, *P < 0.05, **P < 0.01, ****P < 0.0001, as determined by unpaired two-sided Student's t test. P-values are provided for selected comparisons). **c** Representative histological analysis using H&E and IHC staining was performed for Ki-67, pERK, p21, KRAS$^{G12D}$, and Gli1 in orthotopic tumor tissues from 4 mice of each treatment group. Both dtEV and dtEV$^{Gli1}$, combined with GEM, exhibited suppression of Ki-67, pERK, p21, and KRAS$^{G12D}$. Treatment with dtEV$^{Gli1}$ + GEM further reduced Gli1 expression. The siRNA sequences are given in Supplementary Table 3. Source data are provided as a Source data file. (Scale bar in **c**: 100 μm).

## Live cell time-lapsed confocal microscopy

Multi-parametric live cell confocal imaging was operated by an inverted motorized microscope (Nikon TiE, Japan) enclosed by an environmental chamber (OkoLab, Italy). Laser illumination was controlled by iLas2 (GATACA Systems, France) based on Metamorph (Molecular Devices, USA). The measurement and analysis of nanoparticle uptake (vanishing time) on PANC-1 apical cortex and their accumulated numbers are explained in Supplementary Figure 9,10.

### Transwell®-based transcytosis and endocytosis assay

EVs ($1 \times 10^6$ per PANC-1 cell) were first labeled with PKH26 or PKH67 (Sigma-Aldrich). PANC-1 cells were seeded on the porous membrane surface of a Transwell® insert. Once the PANC-1 cells reached 90% confluence, we applied a 1% collagen solution (C5533, Sigma-Aldrich) for 1 h to seal potential leakage between the cells and the transwell before conducting the transcytosis assay. Moreover, to assess the occurrence of leakage, we compared the dtEVs that bypassed the non-cell control (only porous transwell) or PANC-1 cells with or without collagen sealing (Supplementary Fig. 30). The putative transcytosis was determined by the uptake of fluorescent-labeled EVs in cultured PANC-1 cells placed on the bottom surface of the Transwell® insert after 24 h incubation using a Spark® Cyto plate reader (Tecan, Switzerland). To determine the pathways of endocytosis or exocytosis, the PNAC-1 cells on the porous membrane surface were pre-treated with an inhibitor of Pitstop 2 (Pit2, SML1169, Sigma-Aldrich), Methyl-β-cyclodextrin (mβC, C4555, Sigma-Aldrich), Cytochalasin D (CytD, C8273, Sigma-Aldrich), Neticonazole (Ntz, SS717, Selleck), Brefeldin A (BFA, B6542, Sigma-Aldrich), or Exo1 (341220, Sigma-Aldrich) for 24 h before the transcytosis assay.

### Cell viability and cell cycle assay

PANC-1 cells treated with EVs for 24 h were collected and stained with FITC Annexin V Apoptosis/PI Detection Kit (556547, BD Biosciences). For cell cycle assay, the collected cells were first fixed and permeabilized by cold 70% ethanol for 30 min and treated with ribonuclease (20 μg/ml) for 30 min. The permeabilized cells were stained with propidium iodide (50 μg/ml) overnight at 4 °C. Over 10,000 events were captured by a Flow Cytometer (Kaluza, Beckman Coulter, CA, USA) and analyzed by using Kaluza (Beckman Coulter) software.

### Transmission electron microscopy and cryogenic transmission electron microscopy of cells and EVs

Cells collected 4 h after TACE for transmission electron microscopy (TEM) analysis were detached with scraper, washed in PBS, and then fixed in 4% paraformaldehyde solution for 48 h. Samples were treated with 1% osmium tetroxide (OsO4) for 1 h at 4 °C before being dehydrated in increasing grades of alcohol (10 min each: 50%, 70%, 90%; 2 × 5 min: 100%) and embedded in an epoxy resin. Ultrathin sections (70–100 nm) were cut with a Leica Ultracut UCT equipped with a diamond knife (Diatome), transferred to copper grids, stained with 2% aqueous uranyl acetate, and examined with a H-7650 electron microscope at 80 KV. The purified EVs were left on formvar/carbon copper grids, and droplets of EVs were cleared with filter papers. The grids were negatively stained with 2% uranyl acetate for 3 min. After washing with PBS, images were obtained by TEM at 80 KV (H-7650; Hitachi). For EV morphology analysis, 10 μl of EV aliquot sample was applied to a glow-discharged 300-mesh R2.0/2.0 Quantifoil grid. The specimen grid was blotted by Whatman #1 filtration paper and then immediately plunged into liquid ethane to rapidly form a thin layer of amorphous ice by using a Thermo Scientific Vitrobot Mark IV system. The grid was transferred under liquid nitrogen to a Thermo Scientific Glacios Cryo-TEM. Images were recorded by a Thermo Scientific Falcon direct electron detector.

### PANC-1 orthotopic animal studies

In all, 6–8-week-old male NOD/SCID mice were purchased from Shanghai Slack Laboratory Animals in China (IACUC: 2021-0019, approved by Shanghai Model Organisms Center Inc.) or Laboratory Animal Center at Academia Sinica in Taiwan (AS-IACUC-18-03). All animal experiments were approved by the Animal Ethics Committee and experimental procedures were conducted in accordance with the Guidelines of the Institutional Animal Care and Use Committee (IACUC). A small subcostal laparotomy was performed after administration of anesthesia for mice carrying tumor burden of luciferase transduced PANC-1 cells by orthotopically injecting $5 \times 10^5$ (Shanghai, China) or $1 \times 10^6$ (Taiwan) cells. IVIS Spectrum (PerkinElmer, Waltham, America) were measured at 3-week post-implantation. Mice were then randomized for treatment. EVs from MEFs were used and all treatments were administered via intraperitoneal injection three times per week at a dose of $1 \times 10^{11}$ EVs. The Gemcitabine (GEM) dose used was 20 mg/kg with one injection per week. According to the IACUC criteria, individual animal was euthanized if it either lost more than 20% of its body weight or became emaciated in the case of orthotopic mice. Only male mice were utilized in this study to maintain straightforward animal management. Female mice were utilized in other animal studies.

### PANC-1 animal in vivo imaging

NOD/SCID mice were used for in vivo targeting and biodistribution of EVs. Three-week post xenograftment of luciferase/GFP transfected PANC-1 ($1 \times 10^6$ cells), PKH26-labeled liposomes or EVs were injected intraperitoneally. After 24 h, the mice were anaesthetized recorded by IVIS Spectrum. The mice were then euthanized, and major organs were collected for PKH26 fluorescence signal analysis.

### Subcutaneous PDX PDAC animal study

Fresh surgical specimen was obtained from a male patient diagnosed with PDAC (termed T26), who underwent pancreaticoduodenectomy at National Cheng Kung University Hospital (NCKUH) in Taiwan. The study was approved by the Institutional Review Board (IRB number: A-ER-106-157) of NCKUH, and informed consent was obtained from the patient. Whole exome sequencing (WES) of tumor tissue showed both KRAS G12D and TP53[A138V] gene mutations in the cancer cells, and a high ROR1 membrane protein expression on the cell surface by IHC analysis. The tumor specimen from the patient was divided into several fragments and then inoculated into subcutaneous space on the back of 6-week-old female NOD/SCID (Cg-Prkdc[scid]Il2rg[tm1Wjl]/YckNarl) mice purchased from National Laboratory Animal Center of National Applied Research Laboratories in Taiwan. When the tumor size reached approximately $10 \times 10 \times 10$ mm$^3$, the xenograft mouse was sacrificed under anesthesia, and the tumor was excised, trimmed, and cut into small pieces of approximately $3 \times 3 \times 3$ mm$^3$, and subcutaneously inoculated into the mice. At 20-day after tumor transplantation, mice were randomly divided and all treatments were administered via i.p. injection three times a week at a dose of $1 \times 10^{11}$ EVs. The Gemcitabine (GEM) dose used was 20 mg/kg with one injection per week. The tumor volume (V) was calculated as V = length*1/2(width of tumor). All animals were kept in the Laboratory Animal Center under specified pathogen-free conditions at NCKU, with the guide for the animal use protocol approved by the Institutional Animal Care and Use Committee (IACUC number: 110332). Only female mice were utilized in this study to maintain straightforward animal management. The maximum tumor size allowed by IACUC is 20 mm in any direction in an adult mouse and this maximum size was not exceeded.

### Tumor organoids orthotopic PDAC PDX animal study

Fresh surgical specimen was obtained from a patient diagnosed with PDAC (Patient No. PA6247, Crown Bioscience Inc, China). The IRB protocol was approved to collected human sample and informed consent waived. The tumor issue was confirmed by RNA-seq with both KRAS[G12D] and TP53[R196*] gene mutations and high expression levels of Gli1 and ROR1. Mice-bearing tumor organoids (in size of $3 \times 3 \times 3$ mm$^3$) were stitched to the pancreatic tail of 8-week-old male BALB/C-nu mice and confirmed by [18]F-FDG-PET/CT (Inveon Micro-PET/CT, Siemens) measurements at 2-week post-implantation. All animal experiments were approved by the Animal Ethics Committee and experimental procedures were conducted in accordance with the Guidelines of the Institutional Animal Care and Use Committee (IACUC: AN-2104-09-1046, approved by Crown Bioscience Inc.). The dtEVs from hBMSCs were used for treatment in this animal study. All treatments were

administered via i.p. injection 3 times per week at $1 \times 10^{11}$ EVs per dose. Gemcitabine (GEM) was given at a dose of 50 mg/kg once every week. According to the IACUC criteria, individual animal was euthanized if it either lost more than 20% of its body weight or became emaciated in the case of orthotopic mice. Only male mice were utilized in this study to maintain straightforward animal management.

## Micro-PET/CT imaging in orthotopic PDX PDAC mice

PET-CT (Inveon Micro-PET/CT, Siemens) was performed after 2 weeks of inoculation for grouping and another 2 weeks for treatment. BALB/C-nu mice were anesthetized and injected with 2-deoxy-2-[fluorine-18] fluoro-D-glucose ($^{18}$F-FDG) in 100 μl of saline via the tail vein. All mice were kept fasting, with access to water, for 24 h before $^{18}$F-FDG administration and imaging. The mice were maintained under anesthesia and were imaged after 60 min of the injection. Images were reconstructed using an ordered-subset expectation maximization, followed by a maximum a posteriori probability reconstruction algorithm with no attenuation correction and no correction for partial-volume effects. Quantification was performed by volume-of-interest (VOI) analysis using Inveon Research Workplace software (Siemens).

## Histology and immunohistochemistry analysis

All tissues were fixed in formalin and processed for paraffin embedding. Histological analyses on the tumor tissue and other organs, including the liver, lung, heart, spleen, and kidney were performed using H&E staining. For immunostaining, tissue sections were subjected to antigen retrieval. Then the slides were incubated with 0.3% $H_2O_2$ and blocked in 5% bovine serum albumin. The following primary antibodies were used for staining: anti-rabbit p-ERK p44/p42 MAPK (ERK1/2) (Thr202/Tyr204), anti-rabbit Ki-67, Anti-rabbit p21, Anti Human CD64, and RAS (G12D Mutant) antibody.

## Statistics and reproducibility

Data are presented as mean ± SD of triplicates unless otherwise indicated. Statistical analysis was performed using a two-tailed Student's $t$ test or one-way ANOVA with post-hoc tests, as appropriate. For the mouse survival study, the log-rank test was used to test the difference between the two survival curves. The Holm procedure was applied to adjust for multiplicity. The overall family-wise type I error rate was controlled at $\alpha = 0.05$. SAS version 9.3 was used for all statistical analyses (SAS Institute, Inc). For immunoblots and treatment assays, the experiments have been repeated at least three with similar results and representative data was shown.

## Reporting summary

Further information on research design is available in the Nature Portfolio Reporting Summary linked to this article.

## Data availability

The non-coding RNA profiling data in this study are available in the GEO database under the accession code GSE223409. All other data supporting the findings of this study are available within the article and its supplementary files. Any additional requests for information can be directed to the corresponding author (L.J.L.). Source data are provided with this paper.

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

## Acknowledgements

This work was partially supported by The Ministry of Education, Higher Education SPROUT Project for Cancer and Immunology Research Center (112W31101) (L.J.L./C-Y.F.H.), the Yushan Fellow Program (L.J.L.) and NSTC 112-2321-B-A49-019 (C-Y.F.H./Y-S.S./L.J.L.) in Taiwan, National Science and Technology Council (NSTC) and Taiwan Ministry of Health and Welfare (MOHW), grant number NSTC 111-2327-B-006-008, NSTC 112-2321-B-006-010, MOHW112-TDU-B-221-124015 and MOHW112-TDU-B-211-144003 for Y-S.S., Veterans General Hospitals and University System of Taiwan Joint Research Program (VGHUST110-G1-3-2) and Yen Tjing Ling Medical Foundation (CI-110-3) for Y-S.C., and National Natural Science Foundation of China (NSFC 32271401 and NSFC 31971264) and Program of Shanghai Academic/Technology Research Leader in 2022 for Y.Y. We are grateful to Ruey-Hwa Chen. from Academia Sinica in Taiwan for valuable suggestions to this work. We thank A.L. Chun of Science Storylab for critically reading and editing the manuscript.

## Author contributions

L.J.L., C-L.C., and Y.M. designed the study and all experiments with feedback from A.S.L., C-Y.F.H., Y-S.S., M.H., and Y.Y.; C-L.C., Y.M., and J.P. designed and produced ACE/TEP-based dtEVs with help from S-Y.C. and T-S.C.; A.L., Y-S.S., M.H., A.S.L., and R.L. led the animal studies; Y-C.H., Y.M., C-L.C., M-H.C., Z-X.Z., W-H.H., S-Y.C., H-C.L., L.S., and C.Z. conducted animal studies and analysis; C-L.C., Y.M., and J.P. conducted EV characterization with help from X.W., J.Z., H.L., and E.R.; C-L.C., J.P., and Y.M. conducted in vitro experiments; I.L., S.F., and K.W. conducted EV RNA sequencing and analysis; X-Y.C. and Y-S.C. conducted confocal microscopy for single-cell EV endocytosis; L.J.L., C-L.C., Y.M., J.P., and Y-C.H. wrote the paper with feedback from A.S.L., Y-S.S., C-Y.F.H., M.H., Y.Y., K.W., W.J., and B.Y.S.K.

## Competing interests

L.J.L. and A.S.L. are shareholders at Spot Biosystems Ltd. All other co-authors declare no competing interests.

## Additional information

[1]Department of Chemical and Biomolecular Engineering, The Ohio State University, Columbus, OH 43210, USA. [2]Comprehensive Cancer Center, College of Medicine, The Ohio State University, Columbus, OH 43210, USA. [3]Department of Biomedical Engineering, The Ohio State University, Columbus, OH 43210, USA. [4]Institute of Clinical Medicine, College of Medicine, National Cheng Kung University, Tainan 70101, Taiwan. [5]Division of General Surgery, Department of Surgery, National Cheng Kung University Hospital, College of Medicine, National Cheng Kung University, Tainan 70101, Taiwan. [6]Institute of Biopharmaceutical Sciences, National Yang Ming Chiao Tung University, Taipei 11221, Taiwan. [7]Genomics Research Center, Academia Sinica, Taipei 11529, Taiwan. [8]Key Laboratory for Ultrafine Materials of Ministry of Education and School of Materials Science and Engineering, East China University of Science and Technology, Shanghai 200237, PR China. [9]Institute of Systems Biology, Seattle, WA 98109, USA. [10]Brain Research Center, National Yang Ming Chiao Tung University, Taipei 11221, Taiwan. [11]College of Pharmacy, The Ohio State University, Columbus, OH 43210, USA. [12]Department of Radiation Oncology, The University of Texas MD Anderson Cancer Center, Houston, TX 77030, USA. [13]Department of Neurosurgery, The University of Texas MD Anderson Cancer Center, Houston, TX 77030, USA. [14]Institute for Cancer Research, Shenzhen Bay Laboratory, Shenzhen 518055, China. [15]School of Chemical Biology and Biochemistry, Peking University Shenzhen Graduate School, Shenzhen 518055, China. [16]Spot Biosystems Ltd., Palo Alto, CA 94305, USA. [17]These authors contributed equally: Chi-Ling Chiang, Yifan Ma, Ya-Chin Hou. ✉e-mail: cyhuang5@nycu.edu.tw; ysshan@mail.ncku.edu.tw; alee@pku.edu.cn; lee.31@osu.edu

