## [Peer Review File · Nature Communications]

Dual targeted extracellular vesicles regulate oncogenic genes
in advanced pancreatic cancerEditorial Note: Parts of this Peer Review File have been redacted as indicated to remove third-party material where no permission to publish could be obtained.

REVIEWER COMMENTS

Reviewer #1 (Remarks to the Author): with expertise in extracellular vesicle-based delivery systems

In this manuscript, the authors designed and prepared a dual-targeted extracellular vesicle carrying both therapeutic mRNA and siRNA with TACE technology, and received encouraging therapeutic outcomes in PDAC treatment with animal models. Although a decent effort was made in this work, the authors didn't tell a convincing story, and some experimental designs are confusing as well as detailed explanations are required.

Here are some issues that need to be addressed or revised about this project:

1. To our knowledge, the expression of plasmid through electroporation would reach the peak from 24 h to 48 h after transfection. However, the CD64 expression in the study reached the peak 12 h post-transfection, why this would happen?
2. In Figure S8, the authors claimed that the effective transcytosis, but we can only observe the direct evidence of endocytosis or vanish of the labeled dtEVs, it is not actually convincing. And the authors mentioned data from the LNP group was not found, the data is crucial to determine whether the dtEVs were transported to the lower chamber through transcytosis or just leaking.
3. We suggest the authors use intracellular transportation inhibitor Brefeldin A, and exocytosis inhibitor Exo1 to assist in verifying the transcytosis.
4. In Figure 4b, the authors claimed that the dtEVs showed favorable endosomal escape ability, however, the cell at the upper edge revealed the strong colocalization of the dtEVs and the lysosomes.
5. Lots of scale bars are missing, such as in Figure 2b, 4b, 4c, and 7g.
6. The mechanism of the endosomal escape ability should be explained or discussed.
7. Why the siRNA and mRNA cargo be encapsulated separately in the tumor cell inhibition assay? As the release profile difference has been solved through sequential TACE technology.
8. Why the dtEVs were administrated through ip injection but not iv injection, which is much

more common and clinically related?

9. What is the data in Figure 6b? There were so many dots, so what is the sample size (n=?)? And the control data of the LNP group is also missing.

10. Why the statistical analysis was performed between tumors and spleens in Figure 6b?

11. Why the PDX tumors were inoculated subcutaneously? The orthotopic and subcutaneous PDAC pathology is quite different, especially the physiological barriers.

12. Why the type of mice was changed from the NOD/SCID to the BALB/c-Nude? And why the dose of gemcitabine was increased from 20 mg/kg to 50 mg/kg?

Reviewer #2 (Remarks to the Author): with expertise in pancreatic cancer, therapy

This is an interesting and very well conducted manuscript that aims to describe the generation and utilization of dual-targeted extracellular vesicle that carry high load of therapeutic RNA. The efficacy of loaded vesicles was demonstrated using pancreatic cancer as a model. Tissue-specificity was achieved through the integration of a CD64 protein that contains a tissue homing peptide that targets pancreatic tissue. Tumor specificity was ensured by the ROR1 antibody. EV with TP53mRNA or siKRASG12D were produced in MEFs and hBMSCs by Transwell-based asymmetric cell electroporation (TACE). After affinity binding with humanized ROR1 antibodies EVs were applied to orthotopic PDAC models (PANC-1) and PDX to demonstrate the efficacy. The strongest effect was achieved in combination with Gemcitabine.

The experimental set up was very well conducted and the analysis were done with adequate controls. Because this manuscript is mostly directed towards the description of a technology to target genetic alterations in cancer this reviewer has no further comments regarding the generation and application of loaded EV. This has been done very conclusively.

As a use case pancreatic cancer was used in several mouse models that altogether demonstrate the utility and efficacy of loaded EVs.

I do not have specific comments rather than the suggestion to work more on the details of resistencies. In Fig. 6f, the whole-body imaging demonstrates re-growth of the tumor after 49d with dtEV and Gemcitabine. That could be an option to work on some key mechanisms of resistencies.

Several studies have shown that inhibition of oncogenic KRAS in established tumors may

reinforce oncogene-induced senescence (OIS). Thus, it is recommended to stain the tumors with b-Gal and perform key experiments to demonstrate a potential implication of OIS.

Reviewer #3 (Remarks to the Author): with expertise in extracellular vesicles, engineering

The manuscript by Chiang et al. describes development, characterization and functional evaluation of dual targeted extracellular vesicles loaded with both mRNA and siRNA for pancreatic cancer therapy. The work is highly timely and of definite interest to the extracellular vesicle and nanomedicine communities. The conclusions are mainly sound and supported by data. However, particular claims related to loading efficiencies are extraordinary, which in my opinion need to be supported by stronger evidence.

Major comments:

- The TACE procedure for loading plasmid DNA into cells and thereby indirectly boosting EV release and loading is novel. However, it is not clear from the results whether TACE is truly necessary for such effects. Is the observed boost in EV release and RNA loading efficiency specific to the TACE procedure, e.g. when compared to conventional electroporation/nucleofection procedures?
- Can the authors rule out that their EV preparations were contaminated with plasmid DNA, potentially affecting RNA quantification and functional effects?
- Loading efficiency. In order to quantify copy numbers per EV, the authors quantify mRNA/siRNA copies using qPCR and relate this to the total number of EVs as determined by DLS. Using this calculation, they report loading efficiencies for siRNA of up to 1000-1900 copies per EV. This is quite extraordinary given the reported loading efficiencies after plasmid DNA transfection reported by others (see e.g. PMID: 31937944), and the fact that 1900 copies/EV and 1×10^6 EVs/cell would correspond to expression of 1.9×10^9 copies/cell which seems extremely high. Therefore, more evidence to support these claims are required. Firstly, while for qRT-PCR analyses, EVs were isolated using total Exosomes isolation reagent kits, for EV particle number determination, EVs were isolated using a different isolation procedure (although not clearly stated which). For both isolation methods, EV yields and purities are not reported, making it difficult to assess accuracy of the calculations. Ideally, these analyses should be performed on EVs isolated using the same procedure, after demonstrating acceptable purity. Additionally, to demonstrate true, luminal loading, RNA copy numbers after proteinase K and RNase treatment should also be

determined (with and without EV lysis). Finally, did DLS quantification of EV numbers match that of NTA?

- For all in vitro and in vivo studies, both EV and estimated cargo (siRNA/mRNA) doses should be stated.

- Figures 2b and c. Negative controls to demonstrate specificity of the probes are lacking. Details for probes used to stain siRNA are lacking.

- The data to support strong endosomal escape ability (Figure 4) seem less developed. Authors show co-localization on a single time point from which they conclude that because co-localization was lowest for lysosomes endosomal escape must have occurred. However, many other explanations are possible for this observations, including wrong time point, degradation of EVs, or differences in antibody labeling efficiency. In fact, endosomal escape efficiency for LNPs is considered low, while LNP colocalizations show a similar pattern as EVs. I would perhaps recommend to remove this section.

- Can the authors show target engagement (KRAS knockdown using RTqPCR/Western Blot) in vivo?

Minor comments:

- Figure 1e. Can the authors explain the shift in size between synthetic and EV-mRNA?

- Figure 6b. The authors comment on LNPs in the text, however these are not shown in the figure.

- Figure 6c. What does Leu refer to?

- Discussion. A more detailed discussion on the TACE procedure, loading efficiencies and unique targeting properties (In light of other literature) would benefit the reader.

Manuscript NCOMMS-23-18300

Response: We sincerely appreciate the highly constructive comments and valuable suggestions provided by all three reviewers. In response, we have diligently conducted additional experiments, researched relevant literature, and incorporated more comprehensive explanations accompanied by new figures. Furthermore, we have carefully revised certain sections of the manuscript, which have been clearly indicated with **highlighted blue text**. In the subsequent sections, we present a thorough point-by-point response to address each comment raised by the reviewers.

Reviewer #1 (Remarks to the Author): with expertise in extracellular vesicle-based delivery systems

In this manuscript, the authors designed and prepared a dual-targeted extracellular vesicle carrying both therapeutic mRNA and siRNA with TACE technology, and received encouraging therapeutic outcomes in PDAC treatment with animal models. Although a decent effort was made in this work, the authors didn't tell a convincing story, and some experimental designs are confusing as well as detailed explanations are required. Here are some issues that need to be addressed or revised about this project:

We appreciate the constructive comments provided by this reviewer, and as a response, we have conducted additional experiments to specifically address the raised concerns. Furthermore, we have made significant improvements to our revised manuscript by providing a more comprehensive and elucidating explanation of the TACE technology.

1. To our knowledge, the expression of plasmid through electroporation would reach the peak from 24 h to 48 h after transfection. However, the CD64 expression in the study reached the peak 12 h post-transfection, why this would happen?

Response: We express our gratitude to the reviewer for highlighting the distinction between TACE transfection and conventional electroporation techniques. Our results show that TACE delivery, being a non-endocytic method, offers significantly higher efficiency compared to conventional electroporation methods such as bulk electroporation (BEP). In the case of plasmid protein expression, conventional electroporation (i.e., BEP) typically reaches its peak between 24 to 48 hours post-transfection since most plasmids require endocytosis for uptake into transfected cells. BEP involves creating nano-scale pores across the entire cell membrane to facilitate plasmid-membrane binding.

On the other hand, TACE employs a high electric field strength within nanopores, resulting in the formation of temporary pores on localized cell membranes. This unique mechanism allows direct entry of plasmid molecules into the cytosol, bypassing the need for endocytosis. As a result, TACE enables faster protein expression in transfected cells. In our previous study published in *Nature Nanotechnology* in 2011 (Boukany et al., 2011), we demonstrated this phenomenon using a two-dimensional nanochannel chip (NEP) that shares similarities with the TACE chip described in our current manuscript. Referring to **Figure 6c** in that paper (please refer to the *left* side of the following figure), we observed abundant delivery of a reporter gene, GFP plasmid DNA, directly into transfected cells via NEP, whereas some plasmid molecules remained bound to the cell membrane after BEP. Consequently, NEP-transfected cells exhibited strong GFP expression within 6 hours, while BEP-transfected cells required 24 hours to achieve stable GFP expression. We have included this paper in the **References**.

Furthermore, we conducted measurements of human CD64 protein expression in MEF cells at different time

points (6, 12, 18, and 24 hours) following either BEP or TACE transfected. Leveraging the reduced cellular damage associated with TACE, we observed that electroporated cells exhibited CD64 expression as early as 6 hours post TACE electroporation. Conversely, it took over 18 hours for CD64 expression to be observed in cells transfected by BEP. As a result, the released EVs by TACE also expressed CD64 protein earlier compared to those by BEP. We have added the detailed measurement of CD64 expression in transfected MEFs on **Pages 4-5, Lines 108-114**, and included **Supplementary Figure S1** (please refer to the *right* side of the following figure) in our revised manuscript.

[Editorial Note: Nature Nanotechnology is the correct journal name in the left figure below.]

Left, PE Boukany, et al. 2011. *Nature Nanotechnology*

Right, CD64 expression in transfected cells under BEP or TACE.

2. In Figure S8, the authors claimed that the effective transcytosis, but we can only observe the direct evidence of endocytosis or vanish of the labeled dtEVs, it is not actually convincing. And the authors mentioned data from the LNP group was not found.

Response: We apologize for the confusion caused by the misnumbering of **Figures S7-S11**. It was an oversight on our part. To clarify, **Figures S8 and S9** in the original supplementary materials depict a comparison of endocytosis, rather than transcytosis, between LNPs and EVs. The transcytosis comparison between non-targeted LNPs and EVs (Con) is now provided in **Supplementary Figure S10b** of revised manuscript. We have rectified these errors in the revised manuscript, specifically on **Page 8, Line 207**.

the data is crucial to determine whether the dtEVs were transported to the lower chamber through transcytosis or just leaking.

Response: We had checked the possibility leakage. Actually, we grew the PANC-1 cell around 90% confluence and applied 1% collagen solution for 1 h to seal potential leakage between the cells and the transwell before conducting the transcytosis assay. We have provided a more detailed explanation of the Transwell-based endocytosis assay in the **Methods** section and **Supplementary Figure 30** of the revised manuscript. The revisions can be found on **Page 19, Lines 530-544**.

3. We suggest the authors use intracellular transportation inhibitor Brefeldin A, and exocytosis inhibitor Exo1 to assist in verifying the transcytosis.

Response: We appreciate the valuable suggestions provided by the reviewer, as they have contributed to enhancing our understanding of putative transcytosis. To investigate this phenomenon, we conducted experiments employing two proposed inhibitors, namely Brefeldin A (5 μ M) and Exo1 (20 μ M), in our transcytosis system. Through our observations, we have discovered that pretreatment with both Brefeldin A and Exo1 on the first recipient cells led to a decrease in the quantity of dtEVs that underwent transcytosis to the second recipient cells (refer to the figure provided below).

It is widely recognized that both Brefeldin A and Exo1 exert a significant influence on intracellular trafficking between the endoplasmic reticulum (ER) and the Golgi apparatus, which are crucial components for regular exosomal secretion in cells (Islam et al., 2007; Lee et al., 2019). Based on our findings, it is evident that the putative transcytosis process involves the participation of the intracellular vesicular system. However, further research and investigation are necessary to gain a more precise understanding of the underlying mechanism. The new results have been incorporated into **Figure 3f** of the revised manuscript, accompanied by detailed

explanations on Page 8, Lines 214-217.

4. In Figure 4b, the authors claimed that the dtEVs showed favorable endosomal escape ability, however, the cell at the upper edge revealed the strong colocalization of the dtEVs and the lysosomes.

Response: We appreciate the comment raised by the reviewer. Indeed, the images presented in Figure 4b exhibit some edge effects resulting from microscopy. In order to address this concern, we have included three additional replicates in the following figure to provide a more comprehensive demonstration of reduced colocalization and improved endosomal escape ability. These results have been added to the revised manuscript as Figure 4c along with detailed explanations on Page 9, Lines 233-236.

5. Lots of scale bars are missing, such as in Figure 2b, 4b, 4c, and 7g.

Response: We appreciate this comment from the reviewer and have added scale bars on **Figure 2b, 4b, 4c, and 7g** in the revised manuscript.

6. The mechanism of the endosomal escape ability should be explained or discussed.

Response: We appreciate the reviewer for bringing up this significant topic for discussion. Although the detailed mechanism of the endosomal escape ability of extracellular vesicles (EVs) remains unclear, a growing body of evidence supports the idea that EVs' endosomal escape differs substantially from that of conventional liposomal and polymeric nanoparticles. While most synthetic nanoparticle designs achieve endosomal escape by disrupting or destabilizing endosomal compartments (Bonsergent et al., 2021), a recent study by BS Joshi et al. (Joshi et al., 2020) demonstrates that cargo release from EVs within endosomes relies on the interaction of intracellular membranes, rather than endosomal permeabilization.

To investigate this phenomenon, we followed the protocol outlined by BS Joshi et al. and utilized an anti-GFP_nano/fluobody system (a), as depicted in the following figure. In our experimental setup, PANC-1 cells were transfected to express anti-GFP_mCherry fluobody in the cytosol. As the dtEVs carrying CD63^{GFP} (with GFP located in the lumen) underwent endosomal escape, the release of the exosomal lumen induced localized accumulation of anti-GFP_mCherry fluobodies (b). We have provided a detailed description of the design of the anti-GFP_mCherry fluobody used in our study (c). This unique phenomenon of intracellular traffic of EVs, where the integrity of their membrane is maintained until endosomal escape, may also play a crucial role in facilitating the potential transcytosis of EVs.

The process of endosome escape of EVs (as well as other nanocarriers) requires further investigation to better understand its underlying mechanism and quantify its efficacy. We have included the aforementioned figure as **Fig. 4d, 4e and Supplementary Figure 14**, along with explanations in the Results section on **Page 9, Lines 238-250** and the Discussion section on **Page 14, Lines 386-397** of the revised manuscript.

7. Why the siRNA and mRNA cargo be encapsulated separately in the tumor cell inhibition assay? As the release profile difference has been solved through sequential TACE technology.

Response: Since we have already introduced two plasmid DNAs (CD64 and mRNA or siRNA) into the donor cells, we opted to prepare two therapeutic EVs separately to ensure optimal loading of therapeutic RNA for the sake of convenience. However, it is worth noting that our TACE technology has the capability to deliver three or more plasmid DNAs into the donor cells if required.

8. Why the dtEVs were administrated through ip injection but not iv injection, which is much more common and clinically related?

Response: We express our gratitude to the reviewer for inquiring about the intraperitoneal (i.p.) and intravenous (i.v.) injection methods. In general, for long-term experiments requiring multiple injections, i.p. injection is considered more suitable for mice than i.v. injection. It is known that the two lateral tail veins of a mouse can endure approximately six injections (three times on each side), beyond which there is a risk of vessel collapse.

Furthermore, we conducted a small-scale feasibility animal study comparing i.p. and i.v. injections over a short duration, and we did not observe any significant differences in the performance of the administered dtEVs. This circulating concentration of i.p. injection has been provided in the revised manuscript on **Supplementary Figure 20b** and **Page 11, Lines 304-306** to address the question raised by the reviewer.

9. What is the data in Figure 6b? There were so many dots, so what is the sample size (n= ?)? And the control data of the LNP group is also missing.

Response: We apologize for the omission of important information in **Figure 6b**. The biodistribution results depicted in Figure 6b were obtained from mouse organs 24 hours after drug delivery. In the revised manuscript, we have included the animal numbers and specified the LNP group. The animal numbers for each group are as follows: Con_EV: n=4, Flag: n=6, CK_IgG: n=6, CK_EGFR: n=7, CK_ROR1: n=8, LNP: n=4. This information has been provided in **Figure 6b** of the revised manuscript.

10. Why the statistical analysis was performed between tumors and spleens in Figure 6b?

Response: The biodistribution results depicted in **Figure 6b** were obtained from the major organs of mice after a 24-hour period following drug delivery. Since we administered the dtEVs through intraperitoneal (i.p.) injection, we compared the distribution of dtEVs in the tumor with that in the liver or spleen, which served as a background reference. Additionally, it is important to note that the spleen and liver are metabolic organs within the circulatory system.

We appreciate the reviewer's suggestion that a quantitative comparison with the liver would be more appropriate, especially considering the inclusion of LNPs. In response, we have performed a statistical analysis to compare the biodistribution of dtEVs between the pancreatic tumor and the liver. The results of this analysis showing in the following figure have been included in the revised manuscript as the revised **Figure 6b** to address this point.

11. Why the PDX tumors were inoculated subcutaneously? The orthotopic and subcutaneous PDAC pathology is quite different, especially the physiological barriers.

Response: We appreciate the reviewer's attention to the animal models utilized in our study. It is widely recognized that orthotopic patient-derived xenograft (PDX) PDAC animal studies present significant challenges compared to subcutaneous studies. This is primarily due to the requirement of abdominal surgeries in mice to specifically implant the primary PDAC tumor in the pancreatic lobe for the orthotopic model. Overcoming the difficulties associated with the low success rate of the surgical procedure and the potential loss of mice, as well as accurately quantifying tumor size and monitoring changes in orthotopic PDX mice, pose considerable challenges.

To comprehensively address these difficulties, we incorporated both subcutaneous and orthotopic PDX animal models in our study. The subcutaneous PDX model allowed us to assess the efficacy of dtEV treatment, while the orthotopic PDX model provided a practical demonstration of dtEVs' ability to penetrate the physiological barriers of the tumor. We have provided a detailed explanation of this aspect in the revised manuscript, specifically on **Page 12, Lines 335-337**. Additionally, we have included additional tissue staining results obtained from the orthotopic PDX animal study in **Figure 8c**.

12. Why the type of mice was changed from the NOD/SCID to the BALB/c-Nude? And why the dose of gemcitabine was increased from 20 mg/kg to 50 mg/kg?

Response: We appreciate the reviewer's important point regarding the selection of animal models in our study. Choosing the appropriate immunodeficient mouse model to accurately mimic tumorigenesis and treatment response is often a dilemma faced by researchers (Bankert et al., 2001). Initially, we conducted both subcutaneous and orthotopic PDX animal studies using NOD/SCID mice, which exhibit a more comprehensive deficiency in both T cells and B cells. However, we encountered challenges with the fragility of NOD/SCID mice, resulting in poor animal viability and compromised quality after the orthotopic surgery. To address this issue and ensure the feasibility of our study, we made a compromise by utilizing BALB/c-Nude mice for the orthotopic PDX animal study. Although BALB/c-Nude mice only lack functional T cells, they may not be the preferred choice for certain types of animal studies. It is important to note that while the absence of T cells generally results in non-functional B cells, there is still a possibility of accidental activation of existing B cells and NK cells, which could potentially lead to murine immunity resisting the implantation of human PDX tumors and subsequently impacting the evaluation of treatment efficacy.

For the subcutaneous PDX animal study, we continued to use NOD/SCID mice to minimize the potential immune effects from the animals. As a result of using different mouse strains, the suggested doses of the chemodrug, Gemcitabine (GEM) were adjusted accordingly to account for the variations. We have provided a detailed explanation of these considerations in the revised manuscript on **Page 12, Lines 338-340**.

Reviewer #2 (Remarks to the Author): with expertise in pancreatic cancer, therapy

This is an interesting and very well conducted manuscript that aims to describe the generation and utilization of dual-targeted extracellular vesicle that carry high load of therapeutic RNA. The efficacy of loaded vesicles was demonstrated using pancreatic cancer as a model. Tissue-specificity was achieved through the integration of a CD64 protein that contains a tissue homing peptide that targets pancreatic tissue. Tumor specificity was ensured by the ROR1 antibody. EV with TP53mRNA or siKRAS^{G12D} were produced in MEFs and hBMSCs by Transwell-based asymmetric cell electroporation (TACE). After affinity binding with humanized ROR1 antibodies EVs were applied to orthotopic PDAC models (PANC-1) and PDX to demonstrate the efficacy. The strongest effect was achieved in combination with Gemcitabine.

The experimental set up was very well conducted and the analysis were done with adequate controls. Because this manuscript is mostly directed towards the description of a technology to target genetic alterations in cancer this reviewer has no further comments regarding the generation and application of loaded EV. This has been done very conclusively.

As a use case pancreatic cancer was used in several mouse models that altogether demonstrate the utility and efficacy of loaded EVs.

Response: We appreciate the positive feedback by this reviewer.

1. I do not have specific comments rather than the suggestion to work more on the details of resistencies. In Fig. 6f, the whole-body imaging demonstrates re-growth of the tumor after 49d with dtEV and Gemcitabine. That could be an option to work on some key mechanisms of resistencies.

Response: We agree with the reviewer's observation of tumor regrowth after 49 days of treatment. While KRAS and TP53 mutations are prevalent in a majority of PDAC patients and are considered driver oncogenes, targeting KRAS^{G12D} and restoring TP53 alone may not be sufficient to fully sensitize PDAC to chemotherapy. In our PDX animal study (**Figure 7**), we also investigated other important mechanisms of drug resistance, such as GLI1. GLI1, a transcription factor in the Hedgehog signaling pathway, has been implicated in regulating Gemcitabine resistance in pancreatic cancer. By inhibiting the GLI1 pathway through the addition of siGLI1 in dtEVs^{Gli1}, we were able to further reduce orthotopic tumor growth. This highlights the potential of dtEVs carrying multiple genetic cargoes to sensitize PDAC tumors to chemotherapy. However, it's important to note that surgery remains the primary treatment approach for PDAC. Resectable PDAC patients who undergo surgery have reported approximately 25% five-year survival rates, while non-resectable PDAC patients generally have a median survival of less than a year. Our PDX study data supports the potential of dtEVs to sensitize PDAC tumors to Gemcitabine by targeting KRAS^{G12D} and restoring TP53 in an in vivo setting. As a result, non-resectable PDAC patients may have the opportunity to become resectable. We have included this information in the **Discussion** section on **Pages 14-15, Lines 400-412** of our revised manuscript.

2. Several studies have shown that inhibition of oncogenic KRAS in established tumors may reinforce oncogene-induced senescence (OIS). Thus, it is recommended to stain the tumors with b-Gal and perform key experiments to demonstrate a potential implication of OIS.

Response: We appreciate the reviewer's valuable suggestion. Indeed, it is well-established that the aberrant activation of mutant KRAS can override oncogene-induced senescence (OIS), promoting the survival and proliferation of cancer cells (Collado and Serrano, 2010; Lee and Bar-Sagi, 2010). We fully agree with the reviewer's suggestion that the suppression of oncogenic KRAS by our dtEVs could reinforce OIS in tumor

cells. In our *in vivo* studies depicted in **Figure 6h** and **7g-7i**, we observed continuous suppression of KRAS expression in tumor tissues during long-term treatment with dtEVs, accompanied by increased p21 expression and decreased Ki67 levels, indicative of OIS. To further investigate senescence-associated beta-galactosidase (SA- β -gal) activity, we conducted two assays with new results showing in the following figure.

In the first assay, immunohistochemical (IHC) staining of our patient-derived xenograft (PDX) tumors demonstrated the effective suppression of β -gal and Ki-67 expression by dtEVs in primary (a) and metastatic tumor tissues (b). According to the protocol suggested by Debacq-Chainiaux et al. (Debacq-Chainiaux et al., 2009), it is recommended to use freshly prepared samples (frozen slides < 2 hours) for an effective SA- β -gal assay. However, our attempts to measure SA- β -gal activity signals from frozen tissues obtained from previous animal experiments (> 1 year ago) in the second assay did not yield satisfactory results (c). Consequently, we directly conducted the SA- β -gal activity assay on freshly cultured PANC-1 cells to better investigate OIS (d). Following a 48-hour treatment, we observed a robust induction of OIS by dtEVs, both with and without Gemcitabine treatment, as evidenced by an approximately 10-fold increase in SA- β -gal activity compared to the control groups (e) [Scramble: 0.14 ± 0.09 ; GEM: 0.70 ± 0.125 ; dtEV: 1.22 ± 0.35 ; dtEV+GEM: 2.714 ± 38.12]. We have included the detailed measurement of OIS on **Figure 5g, 7h, 7i, Supplementary Figure S18**, and paragraph on **Page12, Lines 329-334** in our revised manuscript.

Reviewer #3 (Remarks to the Author): with expertise in extracellular vesicles, engineering

The manuscript by Chiang et al. describes development, characterization and functional evaluation of dual targeted extracellular vesicles loaded with both mRNA and siRNA for pancreatic cancer therapy. The work is highly timely and of definite interest to the extracellular vesicle and nanomedicine communities. The conclusions are mainly sound and supported by data.

Response: We appreciate the positive feedback provided by this reviewer.

However, particular claims related to loading efficiencies are extraordinary, which in my opinion need to be supported by stronger evidence. Major comments:

1. The TACE procedure for loading plasmid DNA into cells and thereby indirectly boosting EV release and loading is novel. However, it is not clear from the results whether TACE is truly necessary for such effects. Is the observed boost in EV release and RNA loading efficiency specific to the TACE procedure, e.g. when compared to conventional electroporation/nucleofection procedures?

Response: We would like to express our gratitude to the reviewer for raising this important question. We would also like to acknowledge that a similar question was asked by Reviewer#1 regarding the unique phenomenon of TACE in gene expression delivery. TACE-mediated non-endocytic delivery has demonstrated significantly higher efficiency compared to conventional electroporation methods such as bulk electroporation (BEP). With conventional electroporation, protein expression from plasmid DNA typically reaches its peak between 24 to 48 hours after transfection, as most plasmids require endocytic uptake by transfected cells. BEP creates nano-scale pores across the entire cell membrane to facilitate binding of the plasmid to the membrane. In contrast, in TACE, high electric field strength in the nanopores generates larger pores on the cell membrane, equivalent to the pore diameter, allowing plasmid molecules to enter the cytosol directly. This non-endocytic uptake leads to faster protein expression in transfected cells.

We have previously published this phenomenon in our 2011 paper in Nature Nanotechnology (Boukany *et al.*, 2011) using a two-dimensional NEP chip. Additionally, TACE delivers plasmid DNA through nanopores that do not compromise cell viability and enable abundant release of intact extracellular vesicles (EVs) from the cell membrane. This allows for high loading of transcribed therapeutic mRNA (e.g., TP53) or siRNA (e.g., siKRAS^{G12D}, siGLI1), as well as translated membrane proteins (e.g., CD64), within the secreted EVs. The normal lysosome-endosome and autophagy degradation pathways in transfected cells are unable to effectively degrade the unusually large number of transcribed/translated biomolecules, thus contributing to high cargo loading in EVs.

We have provided a detailed explanation of these mechanisms in the **Results (Pages 4-5, Lines 108-114)** and **Discussion (Pages 13, Lines 361-378)** sections of the revised manuscript. In our previous publication (Reference 12 in the revised manuscript), we compared EV secretion and mRNA loading across different cell transfection methods, including conventional electroporation (BEP), NEP (similar to TACE), and lipid nanoparticles (LNP). Our findings demonstrated that NEP (or TACE) was significantly more efficient than BEP and LNP in terms of EV secretion and mRNA loading. Please refer to the accompanying figure for further details. We have included this explanation in the revised manuscript (**Page 5, Lines 137-140**).

[FIGURE REDACTED]

2. Can the authors rule out that their EV preparations were contaminated with plasmid DNA, potentially affecting RNA quantification and functional effects?

Response: We would like to express our appreciation to the reviewer for highlighting this important issue. In our dtEV purification process, we implemented DNase treatment on all collected samples after TACE to eliminate potential contamination of plasmid DNA. To assess the effectiveness of the DNase treatment and measure the plasmid DNA and mRNA contents before and after treatment, we designed a serial qPCR assay. The qPCR assay employed three different primer sets: (i) Exon 10-11 of hTP53 (green, Hs01034249_m1, ThermoFisher), (ii) AmpR (red, Mr00661613_cn, Thermofisher), and (iii) a structural sequence specific to the plasmid DNA (blue). The Exon 10-11 primer set was designed to detect both plasmid DNA and mRNA of TP53 in the sample, while the AmpR and Structure Seq primer sets exclusively target the plasmid DNA. The primer set for plasmid DNA detection was designed to amplify a region spanning 2861 to 3090 bp from the full-length plasmid DNA (4119 bp).

To evaluate the efficacy of DNase treatment, we tested different treatment times (5, 10, or 15 minutes) during the dtEV purification process. Following DNase treatment, EV RNA was extracted, and qPCR analysis was conducted. Our findings demonstrated that a 15-minute DNase treatment effectively removed plasmid DNA without compromising the mRNA content in dtEVs. Less than 0.001% of plasmid DNA, whether AmpR or the structural sequence of the plasmid, was detectable after the 15-minute DNase treatment.

We have included a detailed explanation of the plasmid DNA measurement and removal process on **Page 16, Lines 442-448**, and **Supplementary Figure S26** in the revised manuscript.

3. (part i) Loading efficiency. In order to quantify copy numbers per EV, the authors quantify mRNA/siRNA copies using qPCR and relate this to the total number of EVs as determined by DLS. Using this calculation, they report loading efficiencies for siRNA of up to 1000-1900 copies per EV. This is quite extraordinary given the reported loading efficiencies after plasmid DNA transfection reported by others (see e.g. PMID: 31937944), and the fact that 1900 copies/EV and 1×10^6 EVs/cell would correspond to expression of 1.9×10^9 copies/cell which seems extremely high. Therefore, more evidence to support these claims are required. Firstly, while for qRT-PCR analyses, EVs were isolated using total Exosomes isolation reagent kits, for EV particle number determination, EVs were isolated using a different isolation procedure (although not clearly stated which). For both isolation methods, EV yields and purities are not reported, making it difficult to assess accuracy of the calculations. Ideally, these analyses should be performed on EVs isolated using the same procedure, after demonstrating acceptable purity.

Response: We acknowledge the reviewer's comment on the importance of EV collection, purification, and characterization in the field of EV research. We employ different methods depending on the experimental requirements and limitations. For small-volume EV collection/sorting, such as testing various experimental conditions or time points with limited samples, we utilize the Total Exosomes isolation (TEI) method due to its operational advantages. TEI reagent kits are advantageous for concentrated EV samples with a small volume of less than 3 ml, as it requires a high reagent ratio (culture media: TEI = 2:1). On the other hand, for large-volume EV preparation, particularly for in vitro or animal experiments, we typically employ a tangential flow filtration system (TFF). Although a TFF cycle may take up to 6 hours, it is more affordable and efficient when the sample volume exceeds 10 ml, making it suitable for processing larger sample volumes.

We agree with the reviewer that variations in EV collection, purification, and characterization methods can contribute to inconsistent experimental results. To investigate this, we conducted a direct comparison of the TFF and TEI methods. Following TACE, the culture medium was initially cleared of cell debris and used for the TFF comparison. Half of the purified EV solution was subjected to TFF to estimate the EV yield, while the other half was concentrated using the TEI reagent kit. As shown in the following figure, our findings indicate that the loading efficiency for RNA was similar when using either the TFF or TEI method for qPCR analysis. This explanation has been incorporated into the revised manuscript (**Page 16, Lines 451-452**, and **Supplementary Figure S27**).

Furthermore, we would like to emphasize that we employed size exclusion chromatography (qEV columns) to fractionate EVs into different size ranges in the TFF-purified culture medium. We then measured the number of EVs and RNA loading in each fraction. As depicted in **Figure 2d**, siKRAS^{G12D} was detected in both exosomes and microvesicles (MVs), with a higher number of copies in MVs (5,000-6,000 copies/MV, size:

275.2-461.3 nm), which have a larger volume compared to exosomes (150-250 copies/exosome, size: 61.8-118.0 nm). The siRNA loading levels measured in this independent EV purification and characterization experiment fall within the same range as the average siRNA loading levels obtained from the TFF/TEI isolation methods (**Figure 2d vs. Figure 1c**). Similarly, the measured TP53 mRNA loading levels exhibited a consistent trend between these experiments (**Figure 2d vs. Figure 1c**). Taken together, our study demonstrates that the different EV collection and purification methods employed did not significantly impact the measured RNA loading levels in EVs.

In a study by R Reshke et al. (Reshke et al., 2020) (PMID: 31937944), it was reported that certain cell lines contain abundant microRNAs, reaching approximately 10 copies per small EV (sEV) of around 100 nm. Bulk electroporation on sEVs could further load target siRNA up to 10 times. The total oligo RNA loading in their small EVs is similar to that in our exosomes at size around 100 nm. However, their electroporation was performed on pre-collected sEVs, relying on the passive diffusion of siRNAs into permeable EVs. In our TACE method, we prefer to directly modify the donor cells and collect our dtEVs, which helps maintain the integrity of the dtEV membranes, preventing degradation or potential leakage.

(part ii) Additionally, to demonstrate true, luminal loading, RNA copy numbers after proteinase K and RNase treatment should also be determined (with and without EV lysis)

Response: We appreciate this valuable suggestion provided by the reviewer to enhance the credibility of our findings. For EV RNA quantification, we followed the standard protocol for TEI extraction of highly concentrated extracellular vesicles (EVs) using the ThermoFisher Total Exosome Isolation Kit (4484450). As per the manufacturer's user manual, we consistently treated our samples with proteinase K (0.1-1mg/mL, 37°C, 10 min) in conjunction with the TEI kit. We would also like to emphasize the primary advantage of our TACE system over conventional bulk electroporation, which is the preservation of both cellular and extracellular vesicular membrane integrity, particularly the luminal loading. As suggested by the reviewer, we conducted RNase A treatment (ThermoFisher, EN0531, 5 µg/mL, 30 min at 37°C) with or without permeabilization of the dtEV membrane using 0.3% Triton X-100 (Sigma-Aldrich). As shown in the following figure, when evaluating the loading of TP53 mRNA (**a**) and siKRAS^{G12D} (**b**) in dtEVs, we observed a significant reduction in both TP53 mRNA and siKRAS^{G12D} content when performing RNase treatment accompanied by Triton X-100 permeabilization. The data obtained for TP53 mRNA were as follows: Before: 3.06±0.83; RNase: 2.44±0.33; RNase + Triton: 7.14±0.11x10⁻⁵. For siKRAS^{G12D}, the data were: 1147.0±217.7; 1086.0±196.4; 86.3±48.11. These results indicate that dtEVs with an intact membrane display resistance to degradation. We have incorporated this comparison in the revised manuscript (**Page 16, Lines 459-462, and Supplementary Figure S29**).

(part iii) Finally, did DLS quantification of EV numbers match that of NTA?

Response: We appreciate the reviewer's comment regarding the comparison of NTA and DLS, two commonly used techniques for measuring nanoparticles and exosomes. Each method has its own limitations in terms of detection. NTA requires serial dilution to achieve an appropriate concentration of nanoparticles per frame for tracking Brownian motion, typically aiming for 10-100 nanoparticles. On the other hand, DLS is suitable for nanoparticle concentrations spanning a broader range of 4-5 orders of magnitude.

According to the reviewer's suggestion, we conducted a comparison between NTA (a) and DLS (b) measurements of the dtEVs we collected under the same dilution factor (1:10,000), as shown in the following figure. The concentration measurements obtained by these two methods were found to be comparable, falling within a similar order of magnitude. However, it is important to note that the serial dilution used in NTA may potentially result in the loss of signals from larger particles (>500 nm). Furthermore, DLS exhibits limitations in resolution and sensitivity when it comes to particles smaller than 50 nm.

We have included the comparison of these two methods in **Supplementary Figure S28** and provided an explanation in the revised manuscript to address this point (**Page 16, Line 456-457**).

Dilution	NTA	DLS
1:10	N.A.	$2.83 \pm 0.46 \times 10^{11}$
1:100	N.A.	$4.69 \pm 0.23 \times 10^{10}$
1:1,000	$1.06 \pm 0.06 \times 10^9$	$3.89 \pm 0.21 \times 10^9$
1:10,000	$2.78 \pm 0.07 \times 10^8$	$5.62 \pm 0.13 \times 10^8$
Estimated concentration	$\sim 1.92 \times 10^{12}$	$\sim 4.25 \times 10^{12}$

4. For all in vitro and in vivo studies, both EV and estimated cargo (siRNA/mRNA) doses should be stated.

Response: We have included the estimated RNA cargo in all in vitro studies (**Page 9, Line 259, and Page 10 Lines 264-266**) and *in vivo* studies (**Page 11, Lines 296-297, and 301-302**) in the revised manuscript.

5. Figures 2b and c. Negative controls to demonstrate specificity of the probes are lacking. Details for probes used to stain siRNA are lacking.

Response: We appreciate the reviewer for bringing to our attention the missing information. In **Figure 2b**, the RNA-fluorescence in situ hybridization (FISH) probe sets for each target were obtained from the QIAGEN database. In response, we have included the location and sequence of each probe used for RNA detection as shown in the following figure (**a**). Additionally, we have provided the details of the negative controls for cell staining (**b**) and EV staining, which include (i) the isotype of mouse IgG1, κ (for anti-human CD64 antibody, clone: 10.1, Biolegend); (ii) the isotype of rabbit IgG (for anti-Rab7 antibody, clone: D95F2, Cell Signaling); (iii) a scramble of probes for TP53; and (iv) a scramble of probes for siKRAS^{G12D}. We have tested the specificity of the probes and antibodies with blank EVs, which do not express human CD64, TP53, or siKRAS^{G12D}, as negative controls, as demonstrated in the following figure (**c and d**). The revised manuscript now contains this information on **Page 6, Lines 155-157, and Lines 165-166, and Supplementary Figure S4 and S6**.

6. The data to support strong endosomal escape ability (Figure 4) seem less developed. Authors show co-localization on a single time point from which they conclude that because co-localization was lowest for lysosomes endosomal escape must have occurred. However, many other explanations are possible for these observations, including wrong time point, degradation of EVs, or differences in antibody labeling efficiency. In fact, endosomal escape efficiency for LNPs is considered low, while LNP colocalizations show a similar pattern as EVs. I would perhaps recommend to remove this section.

Response: We appreciate this important comment raised by the reviewer. **Reviewer #1** also raised a similar concern regarding the endosome escape phenomenon. We agree with the reviewer that making multiple comparisons between different endo/lysosomal compartments with various antibodies between dtEVs and LNPs can be complicated. Furthermore, these results alone won't be sufficient to quantify the endosome escape ability of different nanocarriers. To address this concern, we tuned down the endosome escape section and changed the section subtitle from "dtEVs provide strong endosome escape ability" to "Trafficking of dtEVs in recipient cells" and rearranged the colocalization results to avoid the quantitative comparison of any nanocarriers in early endosomes, late endosome, and lysosomes. We revised the first part of this section as follows: "The process of drug delivery within cells and tissues is highly complex. In this study, we present

experimental results aimed at investigating the distinctions in cellular trafficking between dtEVs and other nanocarriers. Typically, recipient cells take up nanocarriers through endocytosis, leading to intracellular trafficking from the endosome to the lysosome (**Fig. 4a**). We treated PANC-1 cells with fluorescent dtEVs, non-targeted IgG-EVs, or LNPs (*indicated by red*) for 2 h. Subsequently, the cells were fixed and co-stained with anti-Rab5 to visualize the extent of colocalization with early endosomes (*green*). We observed similar levels of encapsulation in early endosomes of PANC-1 cells for all three nanocarriers (**Fig. 4b**, co-localization % (Col%): LNP: 89.0 ± 3.5 ; non-targeted EV: 88.17 ± 5.2 ; dtEV: 81.50 ± 3.7 , $n = 3$). In contrast, the levels of colocalization (*indicated by yellow*) among the three nanocarriers in lysosomes exhibited significant variations, with dtEVs showing the lowest level in lysosomes (*indicated by green*) (**Fig. 4c**, Col%: LNP: 26.17 ± 4.0 ; IgG-EV: 13.53 ± 4.87 ; dtEV: 5.73 ± 1.45 , $n = 3$). These findings suggest a potentially superior ability of dtEVs to escape the endosomal compartment. However, further rigorous experimental analyses are required to confirm this observation.”

Furthermore, we conducted additional experiments to gain a clearer understanding of the endosomal escape of dtEVs. Although the detailed mechanism of endosomal escape ability of extracellular vesicles (EVs) remains unclear, there is growing evidence supporting the notion that the endosomal escape of EVs differs significantly from that of conventional liposomal and polymeric nanoparticles. A recent study by BS Joshi et al. (Joshi et al., 2020) demonstrates that the process of cargo release from EVs within endosomes relies on the interaction of intracellular membranes, rather than endosomal permeabilization. To investigate this phenomenon, we followed the protocol outlined by BS Joshi et al., utilizing an anti-GFP_nano/fluobody system, as shown in the following figure (**a**).

In our experimental setup, PANC-1 cells were transfected to express anti-GFP_mCherry fluobody in the cytosol. As the dtEVs carrying CD63^{GFP} (with GFP located in the lumen) underwent endosomal escape, the release of the exosomal lumen induced localized accumulation of anti-GFP_mCherry fluobodies (**b**). We have provided a detailed design of the anti-GFP_mCherry fluobody used in our study (**c**). This unique phenomenon of intracellular traffic of EVs, where the integrity of their membrane is maintained until endosomal escape, may play a crucial role in facilitating the potential transcytosis of EVs as well. The endosome escape of EVs (and other nanocarriers) requires further investigation to better understand its mechanism and quantify its efficacy.

Together, we have revised **Figure 4** and added **Supplementary Figure S14** in the revised manuscript, along with more explanations in the **Results (Page 8-9, Line 224-250)** and **Discussion (Page 14, Line 386-397)** sections.

a

b

c

Anti-GFP_mCherry

```

MDYKDDDDKVLVESGGALVQPGGSLRLSCAASGFPV
NRYSMRWYRQAPGKEREWVAGMSSAGDRSSYEDSV
KGRFTISRDDARNVYVYLMNSLKPEDTAVYYCNVNVG
FEYWGQGTQVTVSSGSGSGSVSKGEEDNMAIKEFMR
FKVHMEGSVNGHEFEIEGEGEGRPYEGTQAKLKVTK
GGPLPFAWDILSPQFMYGSKAYVKHPADIPDYLKLSFPE
GFKWERVMNFEDGGVVTVTQDSSLQDGEFIYKVKLR
GTNFPSDGPMQKKTMGWEASSERMYPEDGALKGE
IKQRLKLDGGHYDAEVKTTYKAKKPVQLPGAYNVNIK
LDITSHNEDYTIVEQYERAEGRHSTGGMDELYK.
    
```

█ Flag █ Linker
█ αGFP_Nanobody █ mCherry

7. Can the authors show target engagement (KRAS knockdown using RTqPCR/Western Blot) *in vivo*?

Response: Yes, we performed Western blotting on the archived tumor tissues obtained from the PDX animal study, as depicted in the provided figure. The knockdown of KRAS expression was validated using anti-KRAS (clone: AT2F8) and anti-KRAS^{G12D} (clone: HL10) antibodies, with a comparison to the scramble control group (SCR). The PANC-1 cell line was utilized as a positive control. We have included the following figure (**Supplementary Figure S21**) and corresponding explanation (**Page 12, Lines 324-325**) in the revised manuscript.

Minor comments:

1. Figure 1e. Can the authors explain the shift in size between synthetic and EV-mRNA?

Response: This is due to the difference in poly-A tail lengths between endogenous mRNAs and synthetic mRNA. Endogenous mRNAs typically have longer poly-A tails with various lengths, usually exceeding 200 bp, whereas synthetic mRNA only adds a 100-120 bp poly-A tail. We have provided an explanation for this difference in the revised manuscript (**Page 5, Lines 132-133**).

2. Figure 6b. The authors comment on LNPs in the text, however these are not shown in the figure.

Response: We apologize for the missing information in **Figure 6b**. The biodistribution results presented in **Figure 6b** correspond to mouse organs collected 24 hours after drug delivery. In the revised manuscript, we have included the animal numbers and specified the LNP group associated with the biodistribution data.

3. Figure 6c. What does Leu refer to?

Response: We apologize for this oversight in our original manuscript. To clarify, "Leu" is the abbreviation used for "Luciferase." In our animal study, PANC-1 cells were transduced with both GFP and luciferase for *in vivo* monitoring. We have made the necessary correction to address this missing information in **Fig. 6c** the revised manuscript.

4. Discussion. A more detailed discussion on the TACE procedure, loading efficiencies and unique targeting properties (In light of other literature) would benefit the reader.

Response: This is an excellent suggestion. We have incorporated additional explanation in the in the **Results section (Pages 4-5, Lines 108-114)** and **Discussion section (Pages 13, Lines 361-378)** of the revised manuscript to provide a more comprehensive understanding of the topic.

References:

- Bankert, R.B., Egilmez, N.K., and Hess, S.D. (2001). Human-SCID mouse chimeric models for the evaluation of anti-cancer therapies. *Trends Immunol* 22, 386-393. 10.1016/s1471-4906(01)01943-3.
- Bonsergent, E., Grisard, E., Buchrieser, J., Schwartz, O., They, C., and Lavieu, G. (2021). Quantitative characterization of extracellular vesicle uptake and content delivery within mammalian cells. *Nat Commun* 12, 1864. 10.1038/s41467-021-22126-y.
- Boukany, P.E., Morss, A., Liao, W.C., Henslee, B., Jung, H., Zhang, X., Yu, B., Wang, X., Wu, Y., Li, L., et al. (2011). Nanochannel electroporation delivers precise amounts of biomolecules into living cells. *Nat Nanotechnol* 6, 747-754. 10.1038/nnano.2011.164.
- Collado, M., and Serrano, M. (2010). Senescence in tumours: evidence from mice and humans. *Nat Rev Cancer* 10, 51-57. 10.1038/nrc2772.
- Debacq-Chainiaux, F., Erusalimsky, J.D., Campisi, J., and Toussaint, O. (2009). Protocols to detect senescence-associated beta-galactosidase (SA-beta-gal) activity, a biomarker of senescent cells in culture and in vivo. *Nat Protoc* 4, 1798-1806. 10.1038/nprot.2009.191.
- Islam, A., Shen, X., Hiroi, T., Moss, J., Vaughan, M., and Levine, S.J. (2007). The brefeldin A-inhibited guanine nucleotide-exchange protein, BIG2, regulates the constitutive release of TNFR1 exosome-like vesicles. *J Biol Chem* 282, 9591-9599. 10.1074/jbc.M607122200.
- Joshi, B.S., de Beer, M.A., Giepmans, B.N.G., and Zuhorn, I.S. (2020). Endocytosis of Extracellular Vesicles and Release of Their Cargo from Endosomes. *ACS Nano* 14, 4444-4455. 10.1021/acsnano.9b10033.
- Lee, K.E., and Bar-Sagi, D. (2010). Oncogenic KRas suppresses inflammation-associated senescence of pancreatic ductal cells. *Cancer Cell* 18, 448-458. 10.1016/j.ccr.2010.10.020.
- Lee, S.S., Won, J.H., Lim, G.J., Han, J., Lee, J.Y., Cho, K.O., and Bae, Y.K. (2019). A novel population of extracellular vesicles smaller than exosomes promotes cell proliferation. *Cell Commun Signal* 17, 95. 10.1186/s12964-019-0401-z.
- Reshke, R., Taylor, J.A., Savard, A., Guo, H., Rhym, L.H., Kowalski, P.S., Trung, M.T., Campbell, C., Little, W., Anderson, D.G., and Gibbings, D. (2020). Reduction of the therapeutic dose of silencing RNA by packaging it in extracellular vesicles via a pre-microRNA backbone. *Nat Biomed Eng* 4, 52-68. 10.1038/s41551-019-0502-4.

REVIEWERS' COMMENTS

Reviewer #2 (Remarks to the Author):

The authors have sufficiently responded to my concerns and added data to clarify the role of oncogene-induced senescence. Also, with regard to resistancy mechanisms important aspects were carefully discussed in the manuscript. Thus, I do not have further comments and would like to commend the authors for this interesting paper.

Reviewer #3 (Remarks to the Author):

The authors have adequately addressed my previous concerns.